# The cell cortex-localized protein CHDP-1 is required for dendritic development and transport in *C. elegans* neurons

Ting Zhao[1,2], Liying Guan[3], Xuehua Ma[3], Baohui Chen[4,5], Mei Ding[3,6]*, Wei Zou[1,2]*

**1** The Fourth Affiliated Hospital, Zhejiang University School of Medicine, Yiwu, China, **2** Institute of Translational Medicine, Zhejiang University, Hangzhou, China, **3** State Key Laboratory of Molecular Developmental Biology, Institute of Genetics and Developmental Biology, Chinese Academy of Sciences, Beijing, China, **4** Department of Cell Biology, and Bone Marrow Transplantation Center of the First Affiliated Hospital, Zhejiang University School of Medicine, Hangzhou, China, **5** Institute of Hematology, Zhejiang University & Zhejiang Engineering Laboratory for Stem Cell and Immunotherapy, Hangzhou, China, **6** University of Chinese Academy of Sciences, Beijing, China

* mding@genetics.ac.cn (MD); zouwei@zju.edu.cn (WZ)

**Data Availability Statement:** All relevant data are within the manuscript and its Supporting Information files.

## Abstract

Cortical actin, a thin layer of actin network underneath the plasma membranes, plays critical roles in numerous processes, such as cell morphogenesis and migration. Neurons often grow highly branched dendrite morphologies, which is crucial for neural circuit assembly. It is still poorly understood how cortical actin assembly is controlled in dendrites and whether it is critical for dendrite development, maintenance and function. In the present study, we find that knock-out of *C. elegans chdp-1*, which encodes a cell cortex-localized protein, causes dendrite formation defects in the larval stages and spontaneous dendrite degeneration in adults. Actin assembly in the dendritic growth cones is significantly reduced in the *chdp-1* mutants. PVD neurons sense muscle contraction and act as proprioceptors. Loss of *chdp-1* abolishes proprioception, which can be rescued by expressing CHDP-1 in the PVD neurons. In the high-ordered branches, loss of *chdp-1* also severely affects the microtubule cytoskeleton assembly, intracellular organelle transport and neuropeptide secretion. Interestingly, knock-out of *sax-1*, which encodes an evolutionary conserved serine/threonine protein kinase, suppresses the defects mentioned above in *chdp-1* mutants. Thus, our findings suggest that CHDP-1 and SAX-1 function in an opposing manner in the multi-dendritic neurons to modulate cortical actin assembly, which is critical for dendrite development, maintenance and function.

## Author summary

Neurons often grow highly-branched cell protrusions called "dendrites" to receive signals from the environment or other neurons. Inside these cells, two types of cytoskeletons, known as the actin cytoskeleton and microtubule cytoskeleton, play essential roles during dendritic branching, growth and function. However, it is not fully understood how the dynamics of the neuronal cytoskeletons are controlled. Using the nematode *C. elegans* (a

**Funding:** This work was supported by the National Natural Science Foundation of China grants (31970919 and 31800861) to W. Z. The funders had no role in study design, data collection and analysis, decision to publish, or preparation of the manuscript.

**Competing interests:** The authors have declared that no competing interests exist.

tiny roundworm found in the soil) as a research model, we found that CHDP-1, a protein localized on the cell cortex, plays a vital role in the formation of actin and microtubule cytoskeleton in the dendrites. Mutations in *chdp-1* cause defective dendrite branching and transport of intracellular organelles. *chdp-1* mutants cannot secrete neuropeptides from the PVD dendrites to module the muscle contraction. Surprisingly, mutating a gene called *sax-1*, which encodes a protein kinase, restores dendrite formation and organelle transport. Our findings reveal novel regulatory mechanisms for dendritic cytoskeleton assembly and intracellular transport.

## Introduction

A neuron is the structural and functional unit of the nervous system in animals. During axon formation and dendrite branching, the plasma membrane of a neuron continuously changes its shape until the morphogenesis process is finished, which heavily relies on the membrane-cytoskeleton interactions [1]. The plasma membrane-associated skeleton, also known as the membrane skeleton, consists of actin, spectrin and associated molecules [2]. Analyses from multiple neuronal cells types from different species revealed that many axons and dendrites contain a specialized periodic actin-spectrin-based membrane skeleton (PMS), which can serve as signaling platforms for RTK transactivation and microtubule maintenance [3–11]. Mutations in spectrin are associated with numerous human diseases, such as hereditary elliptocytosis and spinocerebellar ataxia [12,13]. Loss-of-function mutations in *unc-70/beta-spectrin* in *C. elegans* result in defects in axonal maintenance, cilium biogenesis, neuron migration and dendrite morphogenesis [10,14]. The actin-binding protein, alpha-adducin, has multiple functions in regulating actin cytoskeleton formation. Knock-out of alpha-adducin causes progressive axon enlargement and degeneration [15]. However, compared to what is known for axonal cortical actin assembly, it is still poorly understood how dendritic cortical actin assembly is controlled and whether it is crucial for dendrite development, maintenance and functions.

We previously identified the calponin homology domain-containing protein (CHDP-1) as a critical regulator of cortical actin assembly in *C. elegans*. Loss of *chdp-1* results in defective membrane protrusion formation in the neurite growth cones of BDU and PLM neurons. Using an overexpressed transgene, we showed that CHDP-1 is widely expressed and mainly labels the cell cortex. In BDU and PLM neurons, CHDP-1 promotes cortical actin assembly via recruiting and activating the small GTPase CED-10/Rac1 [16]. It is unclear whether CHDP-1 also regulates cortical actin assembly in multi-dendritic neurons, and if so, whether it is required for dendrite development, maintenance and function.

Here we address these questions using the *C. elegans* PVD neurons as a model. The two PVD neurons, PVDL and PVDR, locate on the left and right sides of the nematode *C. elegans*, respectively, and covers the majority of the surface of the body except for the head and neck regions [17]. These two neurons are born in the middle second larval stage (L2), and each grows an unbranched axon and two primary dendrites (1$^o$) towards the anterior and posterior, respectively. At the late L2 and early third larval stage (L3), secondary dendrites (2$^o$) are formed from the 1$^o$ and grow along the dorsal-ventral axis. When the dendritic tips reach the borders of the outer body wall muscles, the dendrites turn and form T-shaped tertiary branches (3$^o$). At the early L4 stage, quaternary dendrites (4$^o$) are formed from the 3$^o$, and together, they form menorah structures in the wild-type animals [18]. This stereotypical morphogenesis is precisely guided by a multi-protein receptor-ligand complex, including two

transmembrane proteins DMA-1/LRR-TM and HPO-30/Claudin on the dendritic membranes, two transmembrane proteins SAX-7/L1CAM and MNR-1/Menorin on the epidermal membranes, and one secreted protein LECT-2/LECT2 derived from the body wall muscle cells [19–26]. Notably, the high-ordered dendrites are sandwiched between the epidermis and body wall muscles, which is consistent with the role of the PVD neurons as the proprioceptors to sense the contraction of the muscle cells [17,27]. During dendrite development, DMA-1 and HPO-30 promote actin assembly via recruiting/activating TIAM-1 and WRC, respectively [24]. In the PVD dendrites, filamentous actin (F-actin) is enriched in the high-ordered, while the microtubule cytoskeleton is enriched in the $1^o$ dendrites [25,28]. The intracellular organelles, such as the endoplasmic reticulum, mitochondria and secretory/endocytic vesicles, are distributed not only in the primary dendrites but also in the high-ordered ones, which are likely regulated by motor proteins moving along the microtubule cytoskeletons [29,30,31]. In the anterior primary dendrite, a growth cone localized non-centrosomal microtubule organizing center generates plus-end-out microtubules in the growth cone and minus-end-out microtubules along outgrowing dendrites [32]. It is not clear how microtubule assembly is controlled in high-ordered dendritic branches.

In this work, we performed a forward genetic screen and identified a *loss-of-function* mutation in the *chdp-1* gene as the PVD dendrite morphology was defective in the mutant animals. CHDP-1 modulates actin assembly in the dendritic growth cones. Intriguingly, loss of *chdp-1* also perturbs the microtubule cytoskeleton assembly, and transport of intracellular organelles, such as dense-core vesicles, ER and mitochondria in the high-ordered branches. The proprioceptive function of PVD neurons is abolished by the loss of *chdp-1*. Knock-out of *sax-1*, which encodes an evolutionary conserved protein kinase [33,34], rescues the defects mentioned above in *chdp-1* mutants, suggesting that SAX-1 is likely a negative regulator of cortical actin assembly. Together, our results suggest that CHDP-1 and SAX-1 regulate cortical actin assembly, which is critical for proper dendrite development, maintenance and function.

## Results

### Loss of *chdp-1* causes abnormal development of PVD dendrites in *C. elegans*

To identify additional regulators that control dendrite development, we used the multi-dendritic PVD neurons in *C. elegans* as a model and conducted a large-scale forward genetic screen. Among the isolated mutants, here we focused on the characterization of *zac135*, in which the dendrite arborization was significantly affected. Compared to the wild-type controls, the *zac135* mutants showed significantly more $2^o$ and $3^o$ dendritic branches but less $4^o$ branches (Fig 1A and 1B). In addition, the intensity of the membrane-targeted green fluorescent protein (myristoylated-GFP, expressed by an integrated transgene *ser2prom3::myr-gfp*) in the $2^o$ dendritic branches was significantly dimmer in the *zac135* mutants, possibly due to a decreased dendritic width or protein diffusion defect (Fig 1C). Using standard genetic mapping and cloning methods, we identified the causative mutation in *zac135* as a single base change (AT**G** to AT**A**), which disrupted the start codon of CHDP-1 protein (M1I). We also analyzed the dendrite morphology of *tm4947*, a putative molecular null mutant of *chdp-1*[16], and found that both mutants showed similar dendrite branching defects (Fig 1A–1C). Similar abnormal dendrite branching and intensity phenotypes were observed when we used two cytosolic GFP reporters (*wdIs51* and *otIs138*, respectively) to visualize the PVD dendrites [18,35], suggesting that these phenotypes are not specific to the myr-GFP reporter (S1A–S1D Fig). We further analyzed the dendrite width using Stimulated Emission Depletion Microscopy (STED), which offers a higher resolution for imaging [36]. For the $1^o$ dendrites, the width of

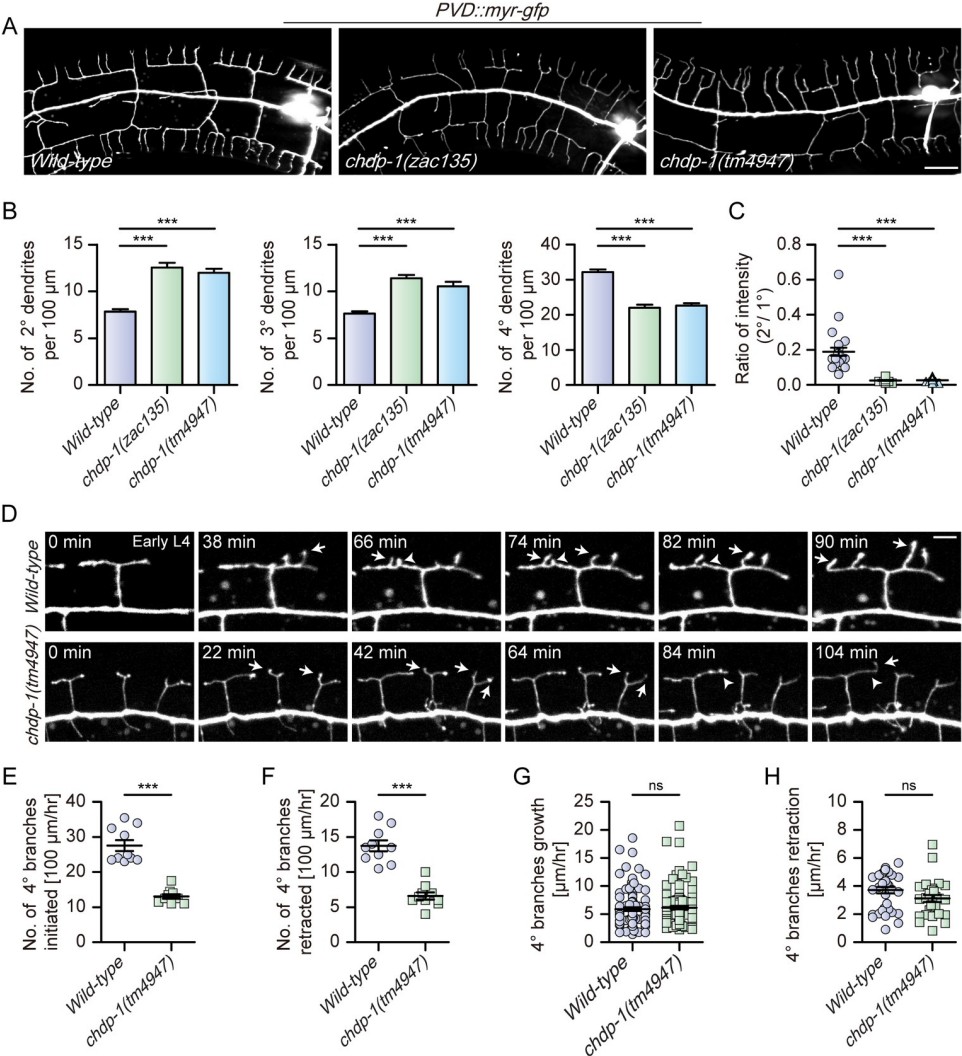

**Fig 1. *chdp-1* mutants are defective in PVD dendrite morphogenesis.** (A) Maximum projection of confocal images showing the PVD morphology of *wild-type*, *chdp-1(zac135)* and *chdp-1(tm4947)* mutant animals at the day 1 adult stage. PVD morphology was visualized using a cell-specific fluorescent marker (*ser2prom3::myr-gfp*). Scale bar: 20 μm. (B-C) Quantification of (B) the number of 2°, 3° and 4° branches, and (C) the ratio of the intensity of the 2° branches to that of the primary dendrite in *wild-type*, *chdp-1(zac135)* and *chdp-1(tm4947)* in the 100 μm area anterior to PVD cell body. Error bars: SEM. ***p < 0.001 by one-way ANOVA with the Tukey correction. n = 20–30 for each genotype. (D) Confocal images from time-lapse movies showing dendrite branching, outgrowth and retraction in *wild-type* (upper) and *chdp-1(tm4947)* mutant animals (lower) during early L4 stage. Arrows: 4° outgrowth, arrowheads: 4° retraction. Scale bar: 5 μm. (E-F) Quantification of the number of 4° branches initiation (E) and retraction (F) per hour in a 100 μm area anterior to the PVD cell body. Error bars: SEM. ***p < 0.001 by Student's t-test. n = 10 animals for each genotype. (G-H) Quantification of the speed of 4° dendrite growth (G) and retraction (H). Error bars: SEM. ns: non-significant by Student's t-test. For G, n = 100 branches for each genotype. For H, n = 36 branches for *wild-type*, and n = 29 branches for *chdp-1(tm4947)*.

*chdp-1* mutants was less uniform than that of the wild-type controls. The average diameters of 1° dendrites are 0.27 μm and 0.34 μm for wild-type and *chdp-1* mutants, respectively. Moreover, the average diameters of 2° branches are 0.17 μm and 0.09 μm for wild-type and *chdp-1* mutants, respectively (S1E–S1G Fig).

To understand why *chdp-1* mutants grow more 2° dendritic branches and less 4° branches, we performed time-lapse recording. We found that *chdp-1* homozygous knock-out mutants

showed more branch initiation and retraction during early larval stage 3 (L3) when $2^o$ dendritic branches were formed. The speed of $2^o$ dendrite outgrowth and retraction were not significantly different between the two genotypes (S2A–S2E Fig). During $4^o$ branch development, *chdp-1* mutants showed less branch initiation and retraction during early larval stage 4 (L4) when $4^o$ dendritic branches were formed. The speed of $4^o$ dendrite outgrowth and retraction were not significantly different between the two genotypes (Fig 1D–1H). Together, our data demonstrate that CHDP-1 plays a vital role in dendrite branch formation.

## Dendrite maintenance is defective in *chdp-1* knock-out animals during the adult stages

To test whether *chdp-1* is required for dendrite maintenance during the adult stages, we examined the dendrite morphologies for both the wild-type control and *chdp-1 (tm4947)* animals at the larval stage 4 (L4), 1 day post L4 stage (day 1), day 3, day 5, day 7 and day 9, respectively. Almost all the animals in the wild-type control groups showed intact anterior primary dendrites from L4 to day 9, while a significant portion of *chdp-1* mutant animals displayed dendrite degeneration in the most anterior part from day 1 to day 9. We quantified the number of $2^o$ and $4^o$ branches in the anterior half of the PVD neurons, roughly between the OLL cell bodies and the vulva and found that the number of $2^o$ branches decreased at day 7 and day 9 when *chdp-1* was mutated (S3A–S3C Fig). Thus, CHDP-1 is required for both dendrite development and maintenance.

## CHDP-1 acts cell-autonomously in the PVD neurons during dendrite branching

Our previous study showed that the exogenously expressed GFP::CHDP-1 driven by the *chdp-1* promoter mainly localizes to the cell cortex in many different cell types [16]. To determine the expression pattern and subcellular localization of endogenous CHDP-1, we inserted the coding sequence of *gfp* into the N-terminus of *chdp-1* locus by CRISPR/Cas9-based genome editing [37]. We quantified the number of dendrite branches of PVD neurons and found that the *gfp*::*chdp-1* knock-in strain and the wild-type control strain showed a similar number of $2^o$, $3^o$ and $4^o$ dendritic branches, suggesting that the *gfp* insertion does not significantly affect the function of CHDP-1 in dendrite development (S4A and S4B Fig). We confirmed that the endogenously expressed CHDP-1 mainly localizes to the cell cortex and is expressed in many cell types, if not all, including the cell bodies of the PVD neurons. Due to the relatively limited resolution of the spinning-disk confocal imaging, we could not determine whether the endogenous CHDP-1 localizes onto the cell cortex in the PVD dendrites (Figs 2A, 2B, S5A and S5B). Next, to determine which tissue CHDP-1 acts in, we expressed CHDP-1 using cell-type specific promoters, including *ser2prom3* for the PVD neurons, *Pdpy-7* for the epidermis, and *Phlh-1* for the body wall muscle cells. *Pchdp-1* was used as a positive control. CHDP-1 expressed from the *chdp-1* endogenous promoter or the PVD promoter fully rescued the dendrite branching defects and the faint staining of $2^o$ branches by the myr-GFP reporter in the *chdp-1(tm4947)* animals, while those driven by the epidermis or the body wall muscle promoter failed to do so (Fig 2C–2E). To understand when CHDP-1 functions, we generated a single copy transgene to express CHDP-1 under a heat-shock promoter (*Phsp-16.48*) in the *chdp-1(tm4947)* genetic background [38]. The increased number of $2^o$ and $3^o$ branches could only be rescued when the transgene expression was induced at the L2 stage, but not at earlier or later stages. The decreased number of $4^o$ branches could be rescued by inducing transgene expression at L2, L3 and L4 stages, but not earlier or later (Fig 2F and 2G). In addition, the faint staining of $2^o$ branches by the myr-GFP reporter in the *chdp-1(tm4947)* animals can be

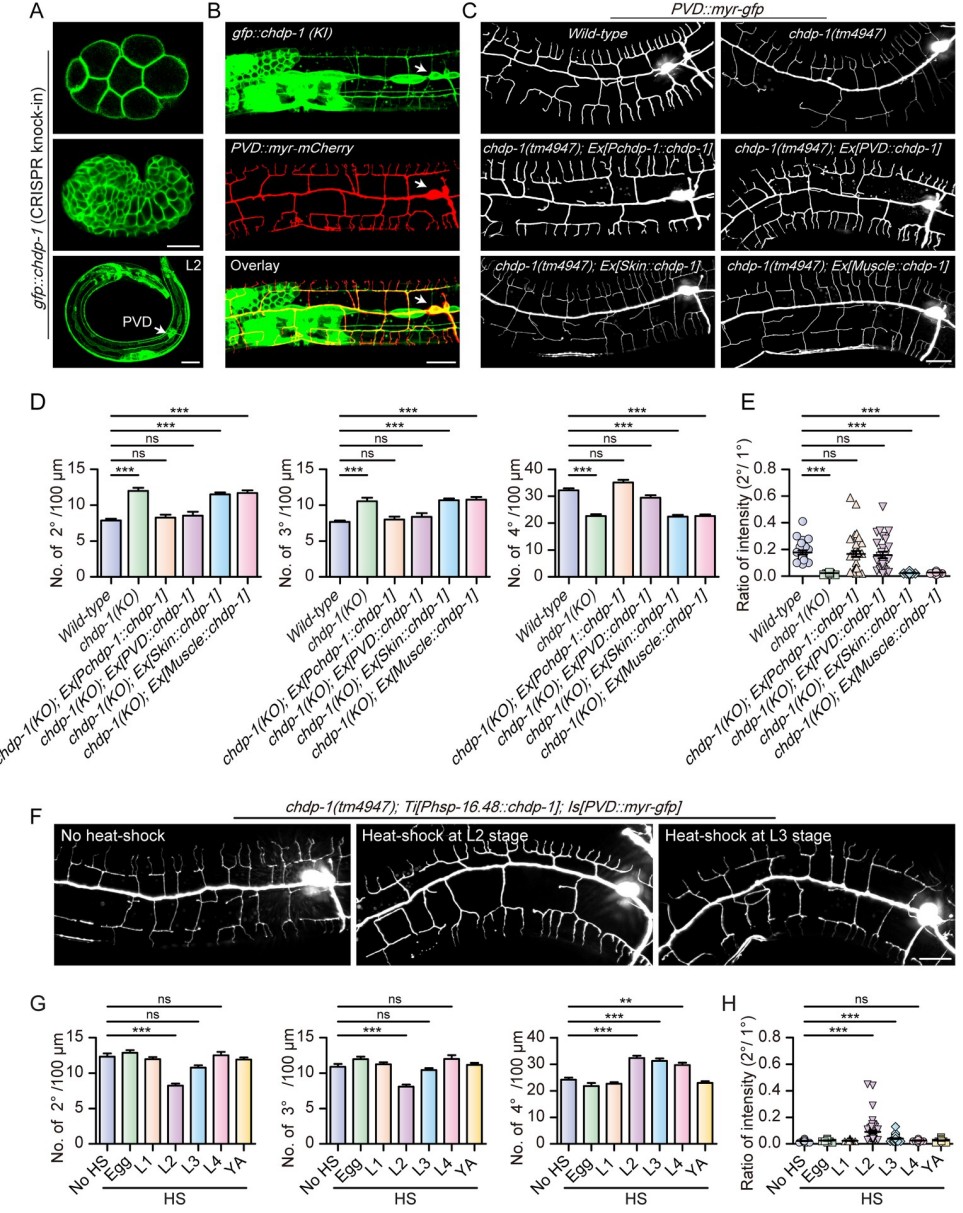

**Fig 2. CHDP-1 functions in the PVD neurons.** Confocal images showing the localization of endogenously expressed GFP::CHDP-1 in embryonic stages (upper, middle) and the second larval stage (lower). Arrows: PVD cell body. Scale bar: 20 μm. (B) Confocal images showing the expression patterns of endogenous GFP::CHDP-1 and myr-mCherry expressed in the PVD neurons (driven by the PVD cell-specific promoter *ser2prom3*). Arrows: PVD cell body. Scale bar: 20 μm. (C) Confocal images of the PVD morphologies in *wild-type*, *chdp-1(tm4947)*, *chdp-1(tm4947)* mutant carrying transgenes driven by *Pchdp-1*, PVD-, skin- and muscle-specific promoters. Scale bar: 20 μm. (D-E) Quantification of (D) the number of 2°, 3° and 4° branches, and (E) the ratio of the intensity of the 2° branches to that of the primary dendrite in *wild-type*, *chdp-1(tm4947)* and *chdp-1(tm4947)* mutant expressing CHDP-1 under different tissue promoters in a 100 μm area anterior to the PVD cell body. Error bars: SEM. ***p < 0.001 by one-way ANOVA with the Tukey correction. ns: non-significant. n = 20–30 for each genotype. (F) Confocal images showing the PVD morphologies of *chdp-1(tm4947)* mutant carrying a *Phsp-16.48::chdp-1* single copy transgene without heat-shock, heat-shocked at either L2 stage or L3 stage. Scale bar: 20 μm. (G-H) Quantification of (G) the number of 2°, 3° and 4° branches in a 100 μm region anterior to the PVD cell body, and (H) the ratio of the intensity of the 2° branches to that of the primary dendrites in *chdp-1(tm4947)* mutant without heat-shock, heat-shocked at different developmental stages. Error bars: SEM. **p < 0.01, ***p < 0.001 by one-way ANOVA with the Tukey correction. ns: non-significant. n = 20–30 for each genotype.

fully or partially rescued by inducing CHDP-1 expression at L2 and L3 stages, respectively (Fig 2H). Together, these data suggest that CHDP-1 functions cell-autonomously and right at the dendrite branching stage.

## Structure-function analysis of CHDP-1 in regulating dendrite morphogenesis

To understand how CHDP-1 regulates dendrite development, we sought to determine which domain(s) is essential by performing the structure-function analysis. The full-length CHDP-1 driven by the PVD promoter fully rescued the dendrite branching defects of *chdp-1(tm4947)* mutants and was used as a positive control. Truncating the P1 motif, P2 motif or the C-terminal part did not affect the rescue ability, suggesting these motifs are not critical for regulating dendrite branching. In contrast, truncating the calponin homology domain (CH) or the helix motif abolished the rescue activity (Fig 3A–3C). We also examined the expression and subcellular localization of the full-length and truncated CHDP-1 proteins tagged by an N-terminal GFP. GFP signals could be detected for all the transgenes. For the full-length, delta P1, delta P2, or delta C transgenes, GFP clearly labeled the cell margin in the cell bodies of the PVD neurons. However, possibly due to improper folding/trafficking, GFP::CHDP-1 delta CH and GFP::CHDP-1 delta helix failed to localize to the cell cortex. They displayed punctate signals and diffused cytosolic distribution, respectively (Fig 3D), consistent with the notion that the cell cortex localization of CHDP-1 is critical for its function in dendrite development.

## Actin assembly in the growth cones of high-ordered branches was reduced by the loss of *chdp-1*

We previously found that CHDP-1 regulates cortical actin assembly during neurite growth cone formation in the BDU and PLM neurons. Thus, we sought to determine whether CHDP-1 plays a similar role during PVD dendrite development. We first examined the localization of the endogenously expressed CHDP-1 in the dendrite growth cones. To avoid the signals derived from other cells, we specifically labeled the endogenously expressed CHDP-1 protein in the PVD neurons using the native and tissue-specific fluorescence (NATF) approach (Fig 4A) [39]. Briefly, we inserted the seven copies of sequences encoding the GFP11 into the N-terminus of the endogenous CHDP-1 using CRIPSR/Cas9-mediated genome editing and overexpressed GFP1-10 specifically in the PVD neurons using the *ser2prom3* promoter from an extrachromosomal array. The animals expressing GFP(7x)::CHDP-1 showed normal dendrite morphology and intensity of the $2^o$ branches, suggesting that the function of the endogenous CHDP-1 is not perturbed (S4C–S4E Fig). Unlike the cytosolic GFP, which showed even distribution in the $2^o$ branches, GFP(7x)::CHDP-1 was enriched in the dendritic growth cones. This pattern was reminiscent of the F-actin probe mCherry::moesin actin-binding domain (moesinABD) (Fig 4B and 4D) [40]. Our previous study reported that the formation of the high-ordered branches relies on F-actin assembly [24]. In the wild-type animals, more than 80% of the moesinABD-labeled growth cones of the $2^o$ branches showed a palm-like shape. In contrast, nearly all the growth cones of the $2^o$ branches in the *chdp-1(tm4947)* animals showed a finger-like shape (Fig 4E–4G). We also compared the actin assembly in the dendritic growth cones of wild-type control and *chdp-1(tm4947)* animals using time-lapse recording and found that knock-out of *chdp-1* dramatically decreased both the growth cone size and F-actin assembly in the growth cones of both $2^o$ and $4^o$ branches (Figs 4G, 4H, S6A, and S6B). The actin assembly defect of the *chdp-1(tm4947)* animals was confirmed using another F-actin reporter —Lifeact::GFP (S7A–S7D Fig) [41].

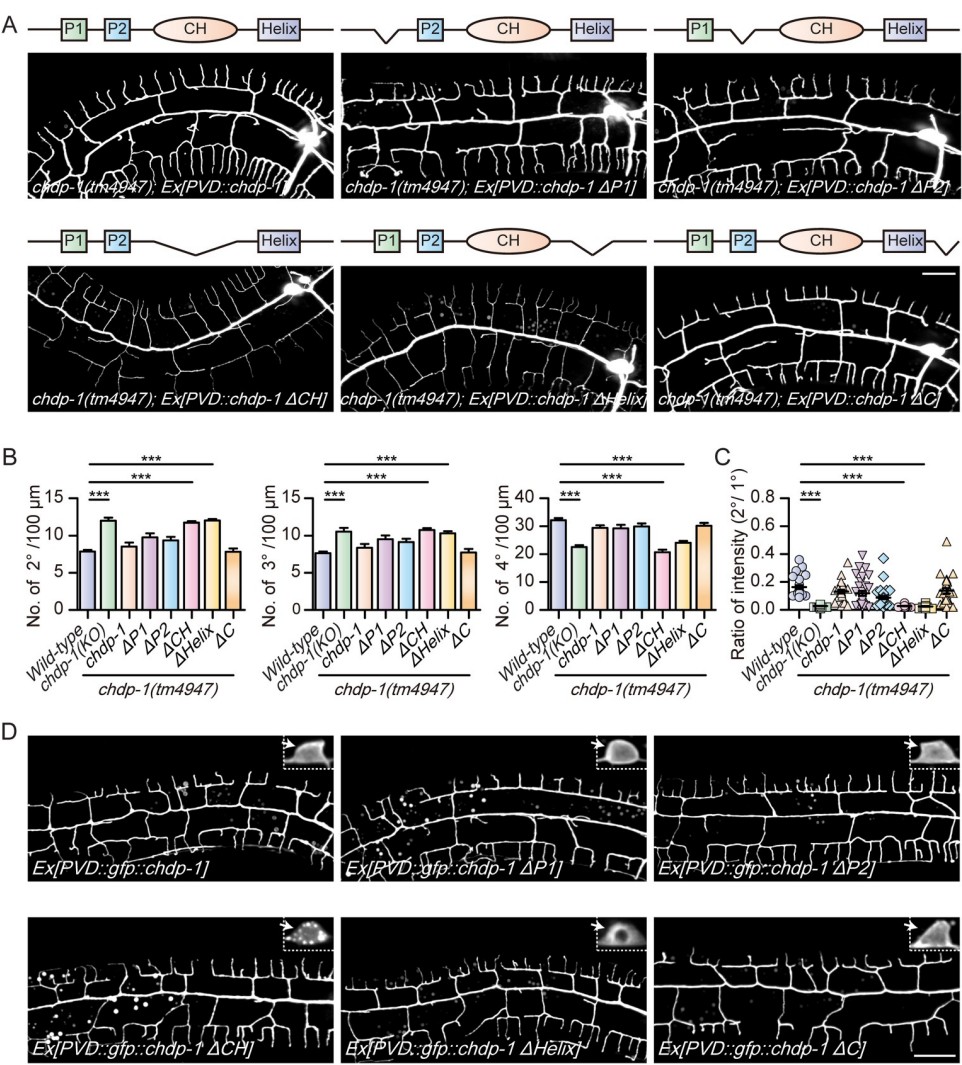

**Fig 3. Structure-function analysis of CHDP-1 in dendrite development.** (A) Confocal images showing the PVD morphologies of *chdp-1(tm4947)* mutant expressing full length CHDP-1, CHDP-1 ΔP1, CHDP-1 ΔP2, CHDP-1 ΔCH, CHDP-1 ΔHelix, CHDP-1 ΔC under the PVD specific promoter. Scale bar: 20 μm. (B-C) Quantification of the number of (B) 2°, 3° and 4° branches in a 100 μm area anterior to the PVD cell body, and (C) the ratio of the intensity of the 2° branches to that of the primary dendrites in *wild-type*, *chdp-1(tm4947)* and *chdp-1(tm4947)* mutant expressing either full length or truncated forms of CHDP-1. Error bars: SEM. ***p < 0.001 by one-way ANOVA with the Tukey correction. n = 20–30 for each genotype. (D) Confocal images showing the localization of GFP::CHDP-1 (full length), GFP::CHDP-1 ΔP1, GFP::CHDP-1 ΔP2, GFP::CHDP-1 ΔCH, GFP::CHDP-1 ΔHelix and GFP::CHDP-1 ΔC under the PVD specific promoter (*ser2rpom3*). Arrows: the localization of CHDP-1 and truncated CHDP-1 in PVD cell body. Scale bar: 20 μm.

## CHDP-1 is required for the proprioceptive function of the PVD neurons

PVD neurons sense the contraction of body wall muscle cells and thus are proprioceptors [17,27]. As loss of *chdp-1* caused defects in dendrite branching and actin cytoskeleton organization, we sought to determine whether CHDP-1 is required for the proprioceptive function of the PVD neurons. We measured the moving tracks and body length of the following four strains: wild-type, *dma-1(wy686)*, *chdp-1(tm4947)* and *chdp-1(tm4947)* carrying a *PVD::chdp-1* transgene. *dma-1* encodes a dendritic branching receptor, loss of which severely affects dendrite branching and the proprioceptive function of the PVD neurons (Fig 5A) [26,27]. Thus,

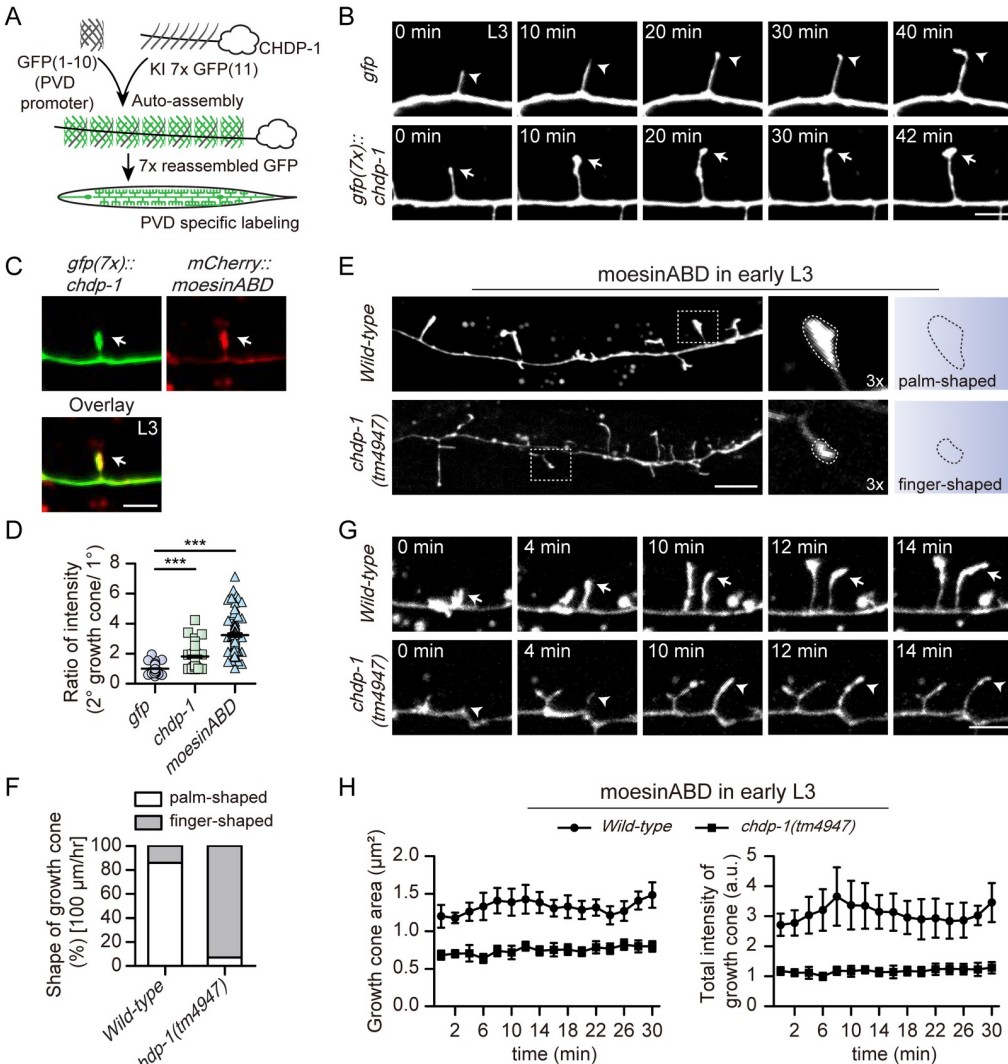

**Fig 4. Loss of *chdp-1* causes reduced actin assembly in the dendritic growth cones.** (A) A schematic diagram illustrating how the endogenously expressed CHDP-1 protein is specifically lighten up by GFP using the native and tissue-specific fluorescence approach. (B) Confocal images from time-lapse movies of growth cones labeled by GFP (upper) and GFP(7x)::CHDP-1 (lower) at L3 stage. Arrowheads: growth cones labeled with GFP, arrows: growth cones labeled with GFP(7x)::CHDP-1. Scale bar: 5 μm. (C) Confocal images showing the localization of GFP(7x)::CHDP-1 (left), mCherry::moesinABD (middle), and overlay (right) at early L3 stage. Arrows: growth cones. Scale bar: 5 μm. (D) Quantification of the ratio of the GFP, CHDP-1 and moesinABD intensity of the 2° dendrite growth cones to that of the primary branches. Error bars, SEM. ***p < 0.001 by one-way ANOVA with the Tukey correction. n = 100 growth cones for each genotype. (E) Confocal images of GFP::moesinABD in *wild-type* (upper), *chdp-1(tm4947)* mutant (lower) at early L3 stage. Scale bar: 10 μm. Dotted line depicts the shape of the developing growth cone. (F) Quantification of the percentage of different morphology of growth cones in *wild-type* and *chdp-1(tm4947)* mutant in a 100 μm area anterior to the PVD cell body. n = 128 2° growth cones (from 10 animals) for WT, and n = 227 2° growth cones (from 10 animals) for *chdp-1* mutants. (G) Confocal images from time-lapse movies of GFP::moesinABD in *wild-type* (upper) and *chdp-1(tm4947)* mutant (lower) during early L3 stage. Arrows: growth cones in *wild-type*, arrowheads: growth cones in *chdp-1(tm4947)*. Scale bar: 5 μm. (H) Quantification of the average area of growth cones labeled by GFP::moesinABD, and the ratio of the intensity of the growth cones in the 2° branches to that of the primary dendrite in *wild-type* and *chdp-1(tm4947)* during early L3 stage. n = 100 growth cones for each group.

*dma-1(wy686)* was used as a positive control in these experiments. All the strains showed similar body lengths, making it possible to directly compare the amplitude and wavelength of the moving tracks (Fig 5B). Interestingly, although *chdp-1(tm4947)* animals grew more dendrite

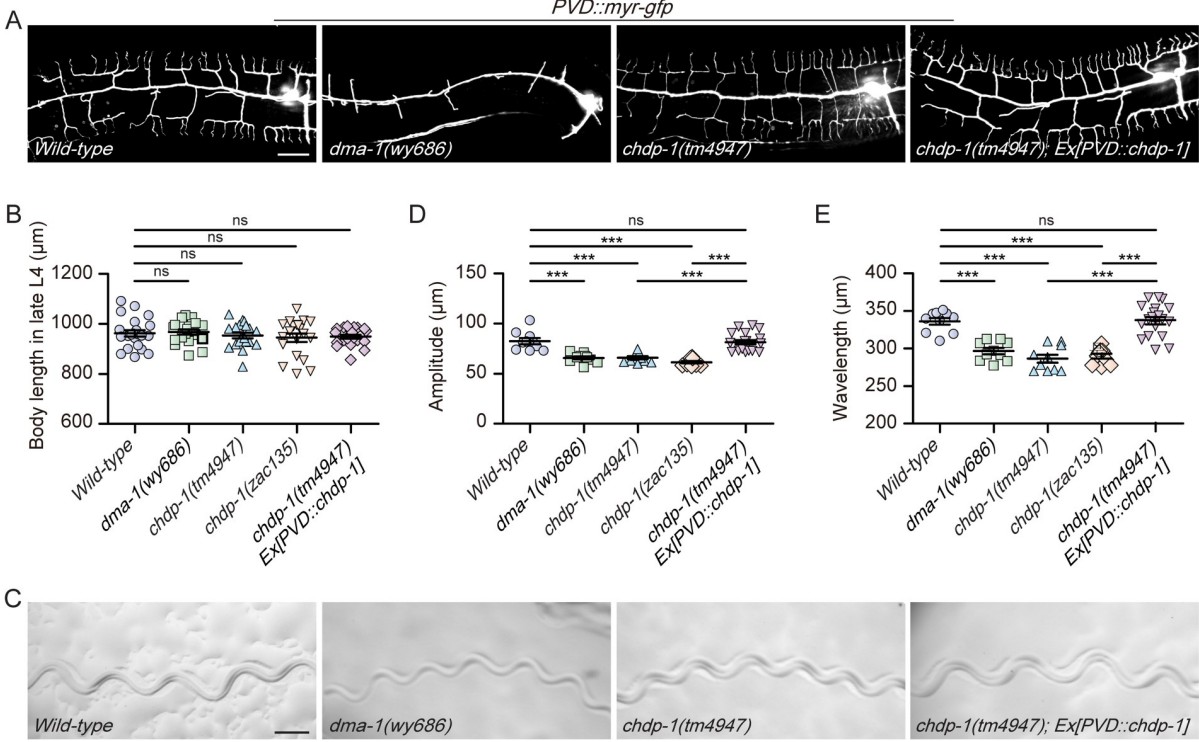

**Fig 5. CHDP-1 is required for PVD's proprioceptive function.** (A) Confocal images of the PVD morphologies of *wild-type*, *dma-1(wy686)*, *chdp-1(tm4947)* and *chdp-1(tm4947)* mutant expressing CHDP-1 under the PVD specific promoter. Scale bar: 20 μm. (B) Quantification of the body length in late L4 stage for the genotypes mentioned above. Error bars: SEM. ns: non-significant. n = 10–20 worms for each genotype. (C) Images showing the representative moving tracks of *wild-type*, *dma-1(wy686)*, *chdp-1(tm4947)* and *chdp-1(tm4947)* mutant expressing CHDP-1 under the PVD neuron-specific promoter. Scale bar: 200 μm. (D-E) Quantification of the (D) amplitude and (E) wavelength of the moving tracks in late L4 stage for the genotypes mentioned above. Error bars: SEM. ***p < 0.001 by one-way ANOVA with the Tukey correction. ns: non-significant. n = 10–20 worms for each genotype.

branches than *dma-1(wy686)* animals, they displayed similar defects in proprioception as determined by the quantification of the amplitude and wavelength of the moving tracks. Expressing CHDP-1 specifically in the PVD neurons fully restored not only the dendrite branching but also the proprioceptive function of the PVD neurons (Fig 5A and 5C–5E). These results reveal that although the high-ordered branches could be generated when *chdp-1* is mutated, they are non-functional for proprioception.

## Neuropeptide release and microtubule assembly are defective in the high-ordered branches of *chdp-1* mutants

Recently Tao et al. reported that the proprioceptive PVD neurons secret neuropeptide NLP-12 from their 3° branches to modulate neuromuscular junction activity and set muscle tone and movement vigor [27]. To understand how the loss of *chdp-1* results in defects in proprioception, we asked whether the dendritic secretion of the neuropeptide NLP-12 is defective in *chdp-1 (tm4947)* animals. We compared the distribution of NLP-12::Venus in wild-type and *chdp-1 (tm4947)* animals. In wild-type animals, NLP-12 positive dense-core vesicles are distributed in the primary dendrites and the high-ordered branches. However, in *chdp-1(tm4947)* animals, these dense-core vesicles can only be found in the primary dendrites (Figs 6A, 6B, S8A, and S8B). Next, to directly compare the local secretion of NLP-12::Venus, we used the neuropeptide trapping assay developed by Tao *et al.* Briefly, the GFP nanobody, which is also known as the

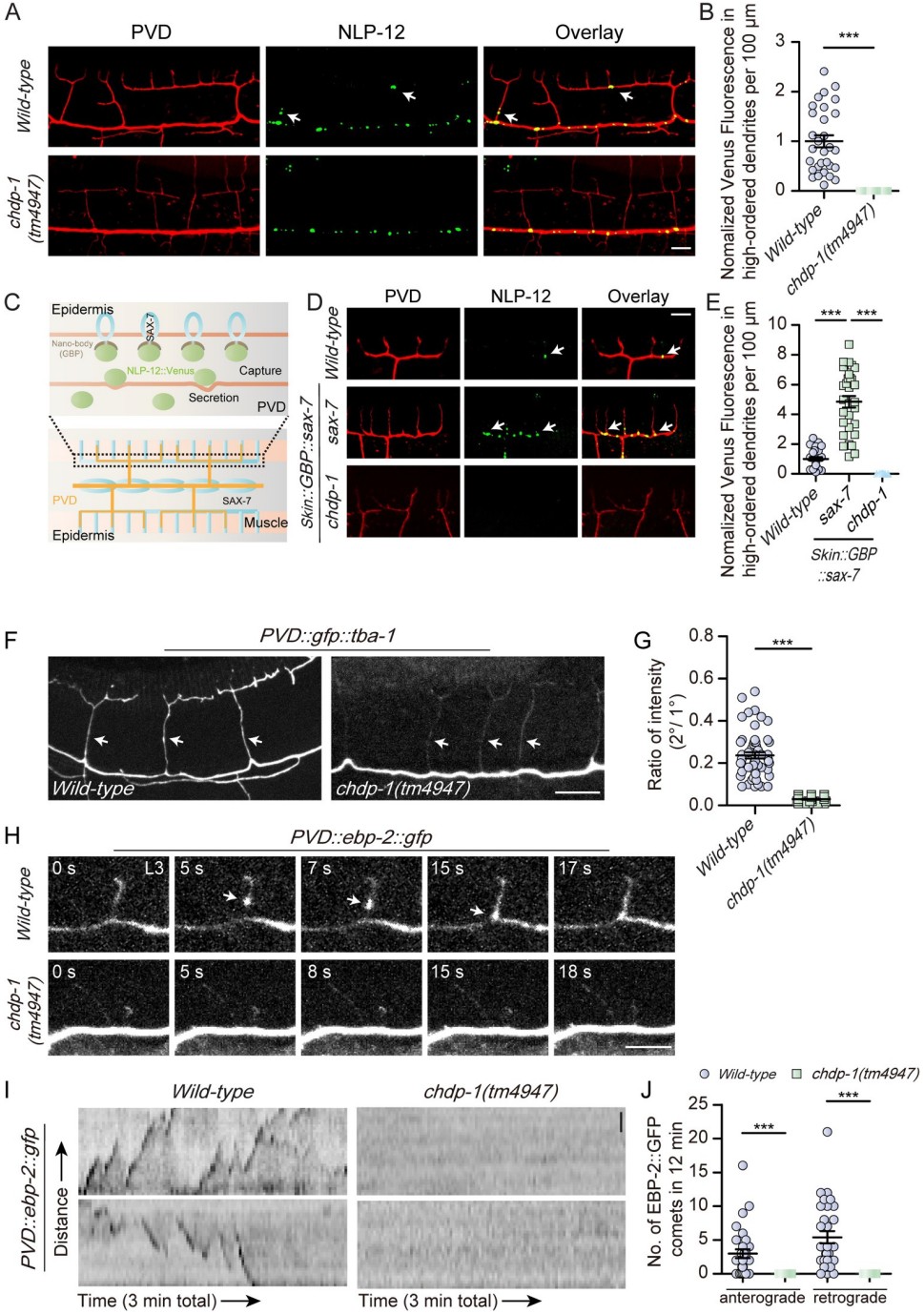

**Fig 6. Loss of *chdp-1* affects assembly of microtubule cytoskeleton and organelle transport in the high-ordered dendrites.** (A) Confocal images of PVD dendrites (left), NLP-12 (NLP-12::Venus) (middle), overlay (right) in *wild-type* (upper), *chdp-1(tm4947)* mutant (lower). Arrows: NLP-12::Venus-positive dense-core vesicles in the high-ordered branches. Scale bar: 20 μm. (B) Quantification of NLP-12::Venus fluorescence intensity in the higher-ordered dendrites in a 100 μm area anterior to the PVD cell body (normalized to WT). Error bars: SEM. ***p < 0.001 by Student's t test. n = 20–30 for each genotype. (C) A cartoon showing the spatial localization of SAX-7 in the epidermis and the rationale of using SAX-7 fused with anti-GFP nanobody (GFP-binding protein, GBP) to capture NLP-12:: Venus secreted from PVD dendrites. (D) Confocal images of PVD dendrites (left), NLP-12 (middle), overlay (right) in *wild-type* (upper), *sax-7(nj48)* (middle), *chdp-1(tm4947)* (lower). Note that both *sax-7(nj48)* and *chdp-1(tm4947)* strains carrying a transgene to express GBP::SAX-7 in the epidermis. Arrows: NLP-12::Venus vesicles in the high-ordered branches. Scale bar: 5 μm. (E) Quantification of NLP-12::Venus fluorescence intensity around the high-ordered dendrites in a 100 μm area anterior to the PVD cell body (normalized to WT). Error bars, SEM. ***p < 0.001

by Student's t test. n = 20–30 for each genotype. (F) Confocal images of GFP::TBA-1 in *wild-type* (left) and *chdp-1 (tm4947)* mutant (right). Arrows: GFP::TBA-1 signals in the 2$^o$ branches. Scale bar: 10 μm. (G) Quantification of the ratio of the intensity of GFP::TBA-1 in the the 2$^o$ branches to that of the primary dendrites in *wild-type* and *chdp-1 (tm4947)*. Error bars: SEM. ***p < 0.001 by Student's t-test. n = 50–60 for each genotype. (H) Confocal images from time-lapse movies showing EBP-2::GFP in PVD neurons in *wild-type* (upper) and *chdp-1(tm4947)* (lower) during early L3 stage. Arrows: a EBP-2::GFP comet moving toward the branching site in a 2$^o$ branch. Scale bar: 5 μm. (I) Kymographs of EBP-2::GFP in secondary dendrites in *wild-type* (left) and *chdp-1(tm4947)* (right) during early L3 stage. Scale bar: 1 μm. (J) Quantification of the number of EBP-2::GFP comets in the secondary branches of *wild-type* and *chdp-1(tm4947)* mutants. The comets move in an either anterograde (away from the 1$^o$/2$^o$ intersection) or retrograde manner (toward the 1$^o$/2$^o$ intersection).

GFP binding protein (GBP), was fused at the N-terminus of the single transmembrane domain protein SAX-7 and the fused protein was specifically expressed in the epidermal cells using the *Pdpy-7* promoter. SAX-7 formed stripes to guide the formation of 3$^o$ and 4$^o$ branches of the PVD neurons. Thus, once NLP-12::Venus was secreted from these branches, the neuropeptide was captured by the locally expressed GBP::SAX-7 due to the high binding affinity between GBP and the Venus protein. We found that loss of *chdp-1* completely abolished NLP-12 secretion from the 3$^o$ dendrites (Fig 6C–6E) [27]. Together, these data suggest CHDP-1 is critical for the dendritic secretion of the neuropeptide NLP-12, and this is likely due to a defect in dense-core vesicle transport from the primary dendrites into the high-ordered branches.

Next, we sought to determine whether loss of *chdp-1* affects the transport of other types of intracellular organelles. The endoplasmic reticulum (ER) distributes in both the primary and some high-ordered dendrites in wild-type animals [31]. However, ER can only be observed in the primary dendrites in the *chdp-1(tm4947)* animals. Dynamic imaging analysis revealed that ER invaded and retracted in some of the 2$^o$ and 3$^o$ branches in the wild-type animals, while no such events was observed in the *chdp-1(tm4947)* animals. Similar defects were also observed for mitochondria (S9A–S9F Fig). We also examined DMA-1::GFP positive vesicles, presumably secretory vesicles and endosomes [30]. In wild-type animals, a small number of DMA-1::GFP positive vesicles could be observed in the high-ordered branches. However, it was extremely difficult for us to identify any vesicles in the high-ordered branches of *chdp-1* mutants. DMA-1::GFP strongly labeled the high-ordered branches in wild-type and *chdp-1 (tm4947)* mutant animals, making it a bit difficult to quantify the number of vesicles. We took advantage of the *exoc-8* mutants, in which the docking/fusion of DMA-1 vesicles onto the dendritic membrane was strongly affected and confirmed that CHDP-1 was indeed required for the transport of the secretory vesicles/endosomes from the primary dendrites into the high-ordered branches (S10A–S10D Fig). Together, these results demonstrate an essential role of CHDP-1 in organelle transport in the high-ordered dendritic branches.

Intracellular organelles are transported by motor proteins, dynein and kinesin, which move along the microtubule cytoskeletons. Thus, we asked whether loss of *chdp-1* affects the assembly of the microtubule cytoskeleton in the PVD dendrites. To examine all the microtubules, which include both the dynamic and stable ones, we generated a GFP::TBA-1 transgene driven by the PVD promoter [25,28]. GFP signals could be observed in the primary and high-ordered dendrites in the wild-type control group. In *chdp-1(tm4947)* animals, GFP::TBA-1 strongly labeled the primary dendrites, while the signal was barely detected in the high-ordered branches (Fig 6F and 6G). To examine the dynamic microtubules, we expressed an EBP-2 (a plus-end binding protein)::GFP reporter in the PVD neurons [28,32]. In both strains, we found a similar number of mobile EBP-2 comets moving either towards the growth cone (anterograde) or the cell body (retrograde) in the growth cones of the anterior primary dendrites. We did not detect any difference regarding the microtubules' running length, growth duration or pause (S11A–S11F Fig). To further test whether there is any microtubule polarity

abnormality, we examined the distribution of the synaptic vesicle maker mCherry::RAB-3. We found no abnormal accumulation of RAB-3 in the dendrites, suggesting that *chdp-1* mutants are not defective in microtubule assembly or polarity in the primary dendrites (S11G Fig). In contrast, we found a small number of EBP-2 comets either moving away from the 1°/2° branching sites (anterograde) or towards the branching sites (retrograde) in the 2° branches of wild-type, but not the *chdp-1(tm4947)* animals (Fig 6H–6J). Thus, in addition to its role in actin cytoskeleton assembly, CHDP-1 is also required for microtubule cytoskeleton assembly in the high-ordered branches.

## CHDP-1 may not function through CED-10

We previously reported that CHDP-1 acts through the small GTPase CED-10/Rac1 in the BDU and PLM neurons [16]. To examine whether CHDP-1 regulates dendritic cortical actin assembly in PVD neurons via a similar mechanism, we examined dendrite morphogenesis in *n3246*, a strong *loss-of-function* allele of *ced-10* [42]. *ced-10 (n3246)* mutants did not show an increased number of 2° and 3° branches or faint staining of the 2° branches by the myr-GFP reporter. The number of 4° branches was reduced in *ced-10 (n3246)* mutants, but to a lesser extent compared to the *chdp-1(tm4947)* animals. In addition, we found that ER distribution in the high-ordered branches was not affected in *ced-10 (n3246)* mutants (S12A–S12E Fig). Together, our data suggest that CHDP-1 regulates dendritic cortical actin assembly via a CED-10-independent pathway or CED-10 only plays a minor role.

## Loss of *sax-1* genetically suppresses defects of *chdp-1* knock-out animals

To understand how dendritic cortical actin assembly is regulated, we searched genes of which *loss-of-function* lead to excess membrane protrusions and thus serve as negative regulators in cortical actin assembly. A previous study reported that loss of *sax-1*, which encodes a conserved serine/threonine protein kinase, causes ectopic membrane protrusion formation [33]. Interestingly, overexpression of CHDP-1 also causes ectopic membrane protrusion [16]. Thus, we built genetic double mutants between *chdp-1(tm4947) and sax-1(ky491)* and examined the genetic interaction between the two genes. Interestingly, loss of *sax-1* fully suppressed the increased number of 2° and 3° branches, decreased number of 4° branches, and faint labeling of myr-GFP in the 2° branches. Loss of *sax-2*, which genetically acts upstream of *sax-1*, also suppressed the defects mentioned above in *chdp-1(tm4947)* (Fig 7A–7C) [34]. Conditional knock-out of *sax-1* in PVD neurons and other cells derived from the seam cell lineage also suppressed these defects in *chdp-1(tm4947)*, indicating that SAX-1 acts cell-autonomously (S13A–S13C Fig). To gain more insights into the suppression, we also analyzed actin assembly in the dendrite growth cones, microtubule assembly and distribution of NLP-12-positive dense-core vesicles in the high-ordered dendrites. Loss of *sax-1* also restored actin assembly, microtubule assembly and dense-core vesicle transport/distribution in the *chdp-1(tm4947)* mutants (Fig 7D and 7E). The double mutant animals also showed nearly normal locomotion: both the amplitude and wavelength were similar to the wild-type control group (S13D–S13G Fig). Together, our results suggest that SAX-1 acts opposingly to CHDP-1, and we speculated that it is a negative regulator of cortical actin assembly.

## Discussion

Here, we reported that the cell cortex-localized protein CHDP-1 acts cell-autonomously in the multi-dendritic PVD neurons to promote cortical actin assembly, which is critical for dendrite formation, maintenance and microtubule-based organelle transport (Fig 8).

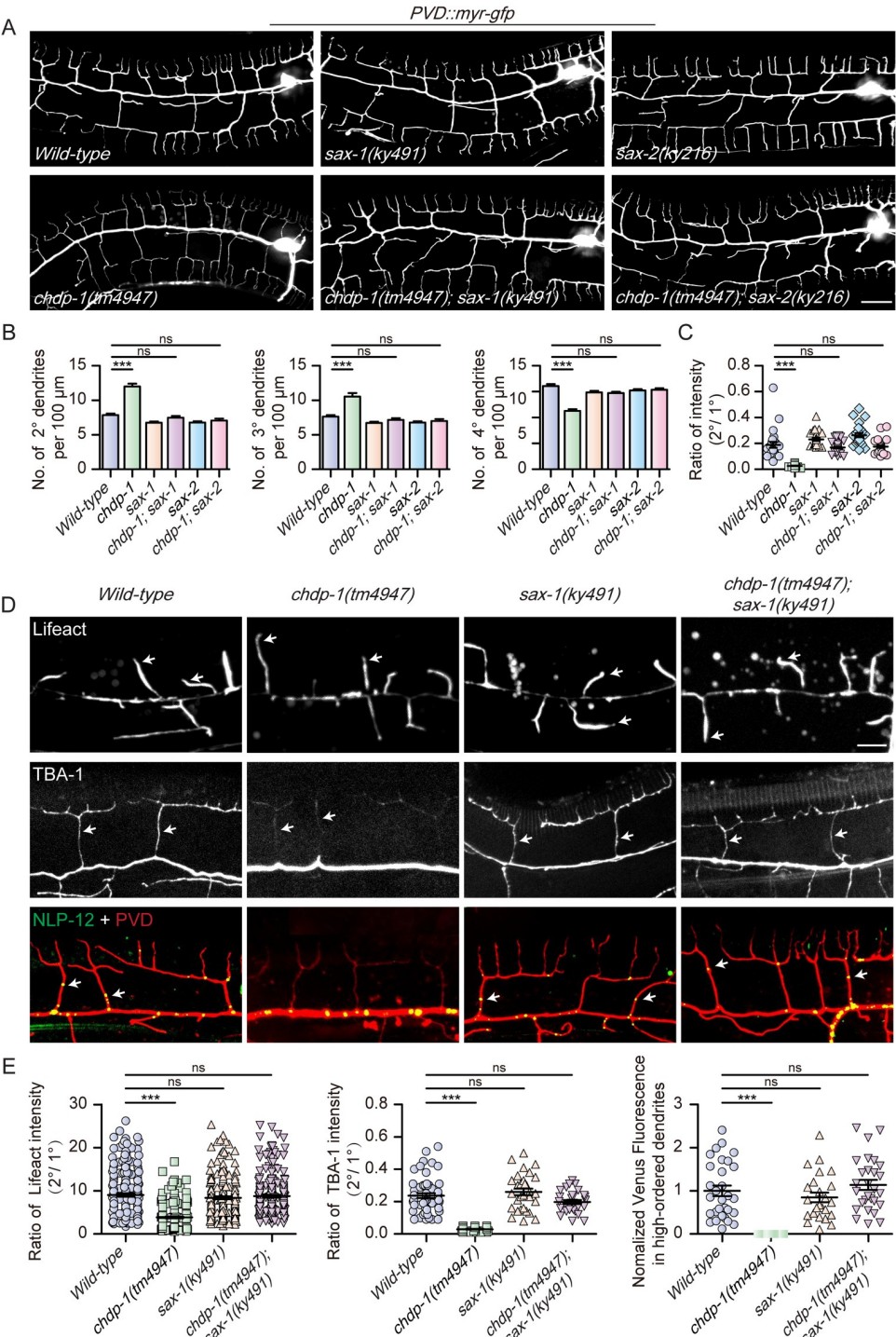

**Fig 7. Knockout of *sax-1* suppresses dendrite development defects in *chdp-1* mutants.** (A) Confocal images of PVD dendrites in *wild-type*, *chdp-1(tm4947)*, *sax-1(ky491)*, *chdp-1(tm4947); sax-1(ky491)*, *sax-2(ky216)* and *chdp-1 (tm4947); sax-2(ky216)*. Scale bar: 20 μm. (B-C) Quantification of (B) the number of 2°, 3° and 4° branches, and (C) the ratio of the 2° branches to that of the intensity of the primary dendrites in the 100 μm area anterior to PVD cell body for the genotypes indicated. Error bars, SEM. ***p < 0.001 by one-way ANOVA with the Tukey correction. ns: non-significant. n = 20–30 for each genotype. (D) Confocal images of (top) Lifeact in early L3 stage (arrows: growth cones of 2° branches with F-actin), (middle) TBA-1 (arrows: 2° branches labeled by GFP::TBA-1), and (bottom) NLP-12::Venus and myr-mCherry (arrows: dense-core vesicles in high-ordered branches) in *wild-type*, *chdp-1(tm4947)*, *sax-1(ky491)* and *chdp-1(tm4947); sax-1(ky491)*. Scale bar: 10 μm. (E) Quantification of (left) the ratio of the Lifeact::GFP

intensity of the 2° dendrite to that of the primary branches, (middle) the ratio of the TBA-1::GFP intensity of the 2° dendrite to that of the primary dendrites, (right) NLP-12::Venus fluorescence intensity in high-ordered dendrites in the 100 μm area anterior to PVD cell body for the genotypes indicated (normalized to WT). Error bars, SEM. ***p < 0.001 by one-way ANOVA with the Tukey correction. ns: non-significant. n = 20–30 for each genotype.

## Dendrite development and maintenance rely on the proper assembly of cortical actin

Neurons are specialized cell types, usually with a long unbranched axon and multiple high-branched dendrites. For dendrites, it has been known for decades that dendrite morphogenesis relies on actin assembly, which is regulated by Rac and Cdc42 [43]. Cortical actin is an enigmatic subset of the actin cytoskeleton. Assembly of cortical actin requires Rac1 and Arp2/3 [16,44,45]. However, these regulators may localize to both cell cortex and cytosol and thus not specific cortical actin regulators. To understand whether cortical actin assembly plays an important role during dendrite morphogenesis and maintenance, a manipulation that disrupts actin assembly at the cell cortex but not anywhere else is required. Our previous study and this study identified that CHDP-1, a cell cortex-localized actin assembly regulator, fulfils this requirement [16]. Through genetic analyses of the putative null mutants of *chdp-1*, we found that loss of *chdp-1* caused an increased number of 2° and 3° branches, and a decreased number

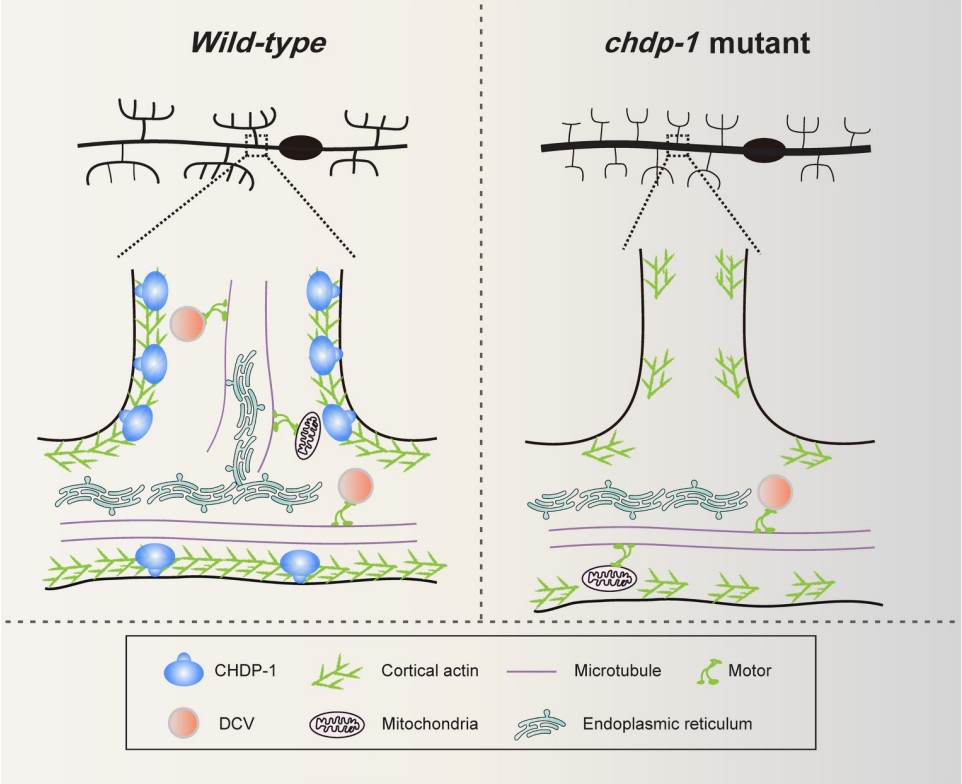

**Fig 8. Working model of CHDP-1 in dendrite development and transport.** A cartoon showing how CHDP-1 regulates cortical actin assembly in the multi-dendritic neurons. In the wild-type animals, CHDP-1 localizes to the cell cortex and promotes cortical actin assembly. In the high-ordered dendrites, CHDP-1-dependent cortical actin assembly is likely required for microtubule assembly and microtubule-based transport. In *chdp-1* knockout animals, cortical actin assembly in the high-ordered branches is reduced, and microtubule assembly and organelle transport are defective. Motor: kinesin or dynein. DCV: dense-core vesicle.

of 4° branches. Disruption of actin assembly has been reported to cause less dendrite branching [24,25]. Thus, it is somehow unexpected that *chdp-1* mutants contain increased 2° and 3° branches. From time-lapse recording, it seems this could be explained by increased 2° initiation (although 2° branch retraction was also observed). Possibly, deceased cortical actin assembly enables more filopodia formation derived from the primary dendrite. However, for the 4° branching, disrupted intracellular transport probably affects branch initiation and stabilization. Using the STED super-resolution microscopy, we also observed opposite defective phenotypes for the diameters of 1° dendrites vs 2° branches: the former was enlarged and the latter was decreased. In addition, the diameter became less uniform for the 1° dendrite in *chdp-1* mutants, reminiscent of the dendrite width in *unc-70/beta-spectrin* mutants [10]. Future study is needed to fully understand why defective cortical actin assembly causes different effects for 1° dendrites and 2° branches. Loss of *chdp-1* also resulted in spontaneous dendrite degeneration in the adults. Interestingly, *loss-of-function* mutations in *unc-70* and alpha-adducin also led to axon degeneration phenotypes [15,46]. Thus, our finding is consistent with the notion that membrane skeleton assembly is critical for neurite maintenance.

## Cortical actin assembly is likely required for microtubule assembly and organelle transport

In the high-ordered branches, loss of *chdp-1* reduced the cortical actin assembly and the microtubule assembly. Two models can explain the decreased microtubule cytoskeleton. (1) CHDP-1 directly acts as a microtubule assembly factor. (2) The effect of CHDP-1 on microtubule assembly is secondary to its function in cortical actin assembly. Actin cytoskeleton often interacts with microtubule cytoskeleton [1,47]. Numerous proteins have been reported to mediate structural interactions between microtubules and actin, such as coronin and Drebrin-EB3 [48,49]. Disrupting of actin cytoskeleton can lead to microtubule assembly defects [50]. Thus, it is conceivable to hypothesize that the cortical actin in the high-ordered branches can serve as a platform to promote microtubule assembly and stabilization. Future studies are needed to tell which model is correct.

## The evolutionarily conserved serine/threonine protein kinase SAX-1/NDR1 is a putative negative regulator of cortical actin assembly

SAX-1 shares high homology with its homolog in multiple species, including CBK1 in yeasts, Trc in flies, and NDR1/2 in humans [51]. Mutants of *sax-1* was initially identified as the cell body of several types of neurons showed ectopic lamellipodia-like protrusions. Disrupting the endogenous RhoA function by expressing a dominant-negative RhoA transgene phenocopied the *sax-1 loss-of-function* phenotype, indicating that SAX-1 might act to modulate actin assembly [33]. Trc is required for proper cell shape and wing hair initiation in flies. Trc mutant cells contained more F-actin than that of the wild-type cells [52]. Knock-out of Trc in the class IV DA neurons results in ectopic dendrite branching and dendritic tilling defects. For dendritic branching, Trc kinase negatively regulates Rac signaling pathway [53]. However, it is unclear whether Rac is a phosphorylation target of Trc kinase. In the present study, we found that loss of *sax-1* suppressed all the defects in the *chdp-1*mutant animals, including abnormal dendrite branching pattern, faint labeling of the 2° branches by the myr-GFP reporter, reduced actin assembly in the growth cones of high-ordered branches, decreased microtubule assembly in the 2° branches, defective organelle transport from the 1° dendrites into the high-ordered branches and impaired proprioception. Very likely, restoration of the cortical actin assembly is the primary effect of *sax-1* knock-out, and other phenotypic restorations are secondary. To the best of our knowledge, this is the first time that the SAX-1/NDR1 kinase family has been

proposed as a putative negative regulator for cortical actin assembly. Currently, the direct phosphorylation target of SAX-1 in this process is not clear. Ultanir *et al*. identified several membrane traffic-related phosphorylated targets for NDR1 kinase using a chemical genetical approach [54]. A future study using a similar strategy will likely uncover the direct downstream player(s) of SAX-1 in the negative regulation of cortical actin assembly.

Together, our results showed that CHDP-1 promotes cortical actin assembly in multi-dendritic neurons. Compromising this process causes defects in dendrite branching, microtubule assembly, organelle transport, neuropeptide secretion and proprioception. Furthermore, our data also suggest that the evolutionarily conserved SAX-1/NDR1 kinase is a putative negative regulator of cortical actin assembly. CHDP-1 and SAX-1 function in an opposing manner to balance cortical actin assembly in neurons and perhaps other non-neuronal cell types. Our study will shed light on the enigmatic mechanisms and functions of neuronal cortical actin assembly.

## Materials and methods

### *C. elegans* genetics

N2 Bristol was used as the wild-type strain. Worms were grown on nematode growth medium plates seeded with OP50 *E. coli* at 20˚C [55]. The mutant alleles used in this study were *chdp-1 (zac135)*, *chdp-1(tm4947)*, *dma-1(wy686)*, *sax-1(ky491)*, *sax-2(ky216)*, *ced-10(n3246)*, *exoc-8 (ok2523)*. For details and complete lists of strains, see S1 Table.

### DNA manipulations and transgenes

Plasmids were constructed by standard methods. pPD49.26 and pSM delta (a derivative of pPD49.26 with additional cloning sites, which was kindly provided by Prof. Kang Shen) were used as vector backbones for most plasmids except the ones for expressing single guide RNAs. Transgenes expressed from extrachromosomal arrays were generated using standard gonad transformation by injection [56]. *Podr-1::rfp*, *Punc-122::rfp*, *Pmyo-2::mcherry*, *Pmyo-3::mcherry* were used as co-injection markers and were injected at 2–50 ng/μl. The single-copy transgene *zacTi22[Phsp-16.48::chdp-1::sl2::bfp]* was generated using the miniMos-based protocol [38]. Briefly, the sequences of *Phsp-16.48*, *chdp-1* cDNA and *sl2::bfp* were amplified from N2 genomic DNA or home-made cDNA library and cloned into pWZ393 (a modified version of pCFJ909, with additional restriction sites and two loxP sites) to generate pZT116. pTZ116 (20 ng/μl), pCFJ601 (*Peft-3::mos1 transposase*, 50 ng/μl), *Pmyo-3::mcherry* (5 ng/μl), *Podr-1::rfp* (50 ng/μl) and *Punc-122::rfp* (50 ng/μl) were injected into the *unc-119(ed4)* animals. Successful single copy transgenes were obtained by screen for animals *non-unc* and without co-injection marker expression. BFP expression was used during the genetic cross. For details and complete lists of plasmids and transgenes, see S1 Table.

### Isolation and mapping of *chdp-1(zac135)* mutants

The *chdp-1(zac135)* mutant was isolated from an F2 semi-clonal screen of 30,000 haploid genomes in *wyIs594* (*ser2p3::myr-gfp* and *Podr-1::rfp*) genetic background. Worms were mutagenized with 50 mM ethyl methanesulfonate (EMS). SNP mapping and whole genome sequencing were performed using standard protocols previously described [57,58]. *zac135* was mapped between Chr I: 0.03 and 2.17. Whole genome sequencing identified five homozygous nonsynonymous mutations in exons of the protein-coding genes (T27A3.5, *chdp-1*, *pdi-3*, F13G3.12 and T25G3.4). Transgenic rescue experiments were carried out, and the results showed that *zac135* was an allele of the *chdp-1* gene.

## CRISPR/Cas9-mediated genome editing

The coding sequence of *gfp* was inserted into the N-terminus of the *chdp-1* locus via CRISPR/Cas9-mediated genome editing [37,39]. Briefly, *Peft-3::cas9+U6::sgRNA for dpy-10* (50 ng/μl, kindly provided by Dr. Suhong Xu), *U6::chdp-1-sgRNA #1* (20 ng/μl, target sequence: 5'AACA-CATCAACT ATGTCTG3'), *U6::chdp-1-sgRNA #2* (20 ng/μl, target sequence: 5'GAGGAAAT-CAAGAAGATCG3'), *U6::chdp-1-sgRNA #3* (20 ng/μl, target sequence: 5'GAGGTCGCCGAGCAAGACA3'), repair template (50 ng/μl) and *Pmyo-2::mcherry* (2 ng/μl) were co-injected into N2. Dumpy or roller F1 animals were picked and cultured for one more generation. Successful knock-in animals were obtained through PCR-based genotyping, and no additional mutation was found based on Sanger sequencing.

Conditional knock-out of *sax-1* in PVD neurons and other cells derived from the seam cell lineage was performed as described previously [30,59]. Briefly, *Pnhr-81::cas9* (50 ng/μl), *U6::sax-1-sgRNA #1* (20 ng/μl, target sequence: 5'GGAAATATCGCAGTACACAA3'), *U6::sax-1-sgRNA #2* (20 ng/μl, target sequence: 5'AAAGCGTGTCACACAATGTG3'), *U6::sax-1-sgRNA #3* (20 ng/μl, target sequence: 5'AGTCTCAAAGTGATTGGACG3'), *Podr-1::rfp* (50 ng/μl) and *Pmyo-2::mcherry* (2 ng/μl) were co-injected into *chdp-1(tm4947); wyIs592* strain. Transgenic lines were obtained by tracking the expression of co-injection markers.

## Confocal imaging of *C. elegans*

For static imaging, worms were immobilized using 1 mM levamisole solution and placed on 4% agarose pads, and then imaged using an Olympus IX83 fluorescence microscope equipped with a spinning-disk confocal scanner (Yokogawa CSU-W1), an sCMOS camera (Prime 95B), and a 60x oil Apochromat objective (NA: 1.49). Z stack images were processed by the projection of maximum intensity except for Figs 2A and S5A.

Time-lapse imaging was performed as previously described with some modifications [60]. Briefly, 2 μl of 1 mM levamisole solution was added into the center of the glass bottom of a microwell dish, then about 20 worms were transferred into the drop of levamisole solution. Next, a 4% agarose pad was gently added onto the animals. All time-lapse movies were taken using the spinning-disk confocal microscope, and Z stack images were processed by projection of maximum intensity except for S10 and S11 Figs, in which single-layer images were shown.

For Stimulated Emission Depletion Microscopy (STED) imaging, worms were immobilized using 1 mM levamisole solution and placed on 4% agarose pads. A Leica TCS SP8 STED fluorescence microscope equipped with 592/660/775 nm lasers, and a HC PL APO CS2 100×/1.40 oil objective was used for imaging. Z stack images were processed by the projection of maximum intensity.

## Split-GFP assay for cell-specific detection of CHDP-1 expression

This assay was performed as described by Siwei He *et al.* [39]. Briefly, the coding sequence for GFP11[7x] was amplified from a previously published plasmid (Addgene #70224). The PCR product was assembled with two homology arms (~ 500 bp each) amplified from the N2 genomic DNA, and the digested pSM delta plasmid as the backbone via the Gibson assembly protocol to generate the repair template plasmid pZT105. A similar CRISPR knock-in strategy was used as how *gfp::chdp-1* was generated, except that pZT105 was used in this experiment. Successful knock-in animals were obtained through PCR-based genotyping, and no additional mutation was found by Sanger sequencing. The coding sequence for GFP1-10 was amplified from a previously published plasmid (Addgene # 70219). The PCR product was cloned into a plasmid in which the PVD-specific promoter *ser2prom3* was previously inserted. This plasmid (pZT106, 20 ng/μl), *Podr-1::rfp* (50 ng/μl) and *Pmyo-2::mcherry* (2 ng/μl) were injected into the

*zac283[gfp11(7x)::chdp-1]* strain. Stable transgenic lines were isolated via the expression of the co-injection markers and subjected to confocal imaging.

### Local neuropeptide secretion assay

This assay was performed as described by Li Tao et al. [27]. Briefly, pZT143 *ser2rom3*::*nlp-12*::*venus*::*sl2*::*mcherry* (20 ng/μl), pLT110 *Pdpy-7*::*gbp*::*sax-7* (10 ng/μl, kindly provided by Dr. Kang Shen), *Pmyo-2*::*mcherry* (2 ng/μl) and *Podr-1*::*rfp* (50 ng/μl) were injected into *sax-7 (nj48)* or *chdp-1 (tm4947)* mutant animals. Stable transgenic lines were obtained and subjected to confocal imaging.

### Locomotion assay

This assay was performed following a previous protocol with some modifications [61]. Briefly, 10–20 worms in the late L4 stage were transferred individually into fresh NGM plates, and then the plates were put in a 20˚C incubator for 1–2 hours. Images of the crawl tracks were taken using a Nikon SM218 stereo microscope with a 1x SHR Plan Apo objective (NA: 0.15). The trajectory's amplitude (the distance between opposite peaks) and wavelength (the distance between two successive peaks) were measured using ImageJ. For each strain, the crawling trajectories of 10–20 worms were measured.

### Quantification and statistical analysis

For PVD dendrites, the number of dendrites of 2˚, 3˚, and 4˚ is the number within the 100 μm region anterior to the PVD cell body. For the fluorescence intensity and width of dendrites, ImageJ is used for statistics. The numerical data that underlies graphs or summary statistics were summarized in S2 Table. The Student's t-test (for the difference between two groups) or one-way analysis of variance with Tukey correction (for the difference between three or more groups) was used for statistical analysis.

### Supporting information

**S1 Fig. Loss of *chdp-1* causes PVD dendrite branching defects.** (A) Confocal images showing the PVD morphologies of *wild-type* (left), *chdp-1(zac135)* (middle) and *chdp-1(tm4947)* (right) animals labeled using *wdIs51* which expresses cytosolic GFP driven by the F49H12.4 promoter in PVD neurons and a few other neurons. Scale bar: 20 μm. (B) Quantification of the number of 2°, 3° and 4° branches in a 100 μm area anterior to the PVD cell body, and the ratio of the intensity in the 2° branches to that of the primary dendrites in *wild-type*, *chdp-1(zac135)* and *chdp-1(tm4947)*. The dendrites are labeled using *wdIs51*. Error bars, SEM. ***p < 0.001 by one-way ANOVA with the Tukey correction. n = 20–30 for each genotype. (C) Confocal images showing the PVD morphologies of *wild-type* (left), *chdp-1(zac135)* (middle) and *chdp-1(tm4947)* (right) animals labeled using *otIs138*, which expresses cytosolic GFP driven by the *ser2prom3* promoter (the expression level is much lower than that of *wyIs592*). Scale bar: 20 μm. (D) Quantification of the number of 2°, 3° and 4° branches in a 100 μm area anterior to the PVD cell body, and the ratio of the intensity in the 2° branches to that of the primary dendrites in *wild-type*, *chdp-1(zac135)* and *chdp-1(tm4947)*. The dendrites are labeled using *otIs138*. Error bars, SEM. ***p < 0.001 by one-way ANOVA with the Tukey correction. n = 20–30 for each genotype. (E) Images taken by the STED super-resolution microscopy for primary dendrites (left) and secondary dendrites (right) in *wild-type* and *chdp-1(tm4947)*. Note that the orientation for the 2° branches is as following: left: close to the 1° dendrite. Right: close to the 3° branches. Scale bar: 500 nm. (F) Plots of the width along the

representative primary dendrite and 2° branches in a 10 μm area. (G) Quantification of the mean width of the primary dendrite and 2° branches in *wild-type* and *chdp-1(tm4947)*. Error bars: SEM. ***p < 0.001 by Student's t test. n = 20 for each genotype.
(TIFF)

**S2 Fig. Time-lapse analyses of dendrite formation in *wild-type* and *chdp-1* mutant animals in early L3 stage.** (A) Confocal images from time-lapse movies showing dendrite branching, outgrowth and retraction in *wild-type* (upper) and *chdp-1(tm4947)* mutant animals (lower) during early L3 stage. Arrows: 2° outgrowth, arrowheads: 2° retraction. Scale bar: 5 μm. (B-C) Quantification of the number of 2° branches initiation (B) and retraction (C) per hour in a 100 μm area anterior to the PVD cell body. Error bars: SEM. ***p < 0.001 by Student's t-test. n = 10 animals for each genotype. (D-E) Quantification of the speed of 2° dendrite growth (D) and retraction (E). Error bars: SEM. ns: non-significant by Student's t-test. For D, n = 100 branches for each genotype. For E, n = 36 branches for *wild-type*, and n = 29 branches for *chdp-1(tm4947)*.
(TIF)

**S3 Fig. Loss of *chdp-1* causes an age-dependent dendrite degeneration in adult animals.** (A) A cartoon showing the simplified morphologies and locations of OLL neurons in the head region and PVD neurons. Area is defined to the distance between the cell bodies of OLL and vulva. (B) Confocal images of the PVD dendrites in *wild-type* and *chdp-1(tm4947)* mutants at the fourth larval, 1 day-old adult (DOA), 5 DOA, 9 DOA stages. Arrows: the distal tip of the anterior primary dendrite. Asterisks: cell bodies of OLL neurons. Scale bar: 50 μm. (C) Quantification of the number of 2° and 4° branches per area in *wild-type* and *chdp-1(tm4947)* mutant at six different developmental stages, including L4, 1 DOA, 3 DOA, 5 DOA, 7 DOA and 9 DOA. Error bars: SEM. *p < 0.05, ***p < 0.001 by one-way ANOVA with the Tukey correction. ns: non-significant. n = 50–60 animals for each group.
(TIF)

**S4 Fig. GFP knock-in does not affect the function of endogenous CHDP-1.** (A-B) Quantification of the number of (A) 2°, 3° and 4° branches in a 100 μm area anterior to the PVD cell body, and (B) the ratio of the intensity in the 2° branches to that of the primary dendrites in *wild-type* and *gfp*::*chdp-1* knock-in animals. Error bars: SEM. ns: non-significant by Student's t-test. n = 20–30 for each genotype. (C) Confocal images showing the expression patterns of GFP(7x)::CHDP-1 (left), mCherry (middle) and overlay (right) in the PVD neurons. Scale bar: 20 μm. (D-E) Quantification of (D) the number of 2°, 3° and 4° branches in a 100 μm area anterior to the PVD cell body, and (E) the ratio of the intensity of the 2° branches to that of the primary dendrite in *wild-type* and *gfp(7x)*::*chdp-1* knock-in animals. Error bars: SEM. ns: non-significant by Student's t-test. n = 20–30 for each genotype.
(TIF)

**S5 Fig. Endogenous CHDP-1 localizes to the cell cortex.** (A) Left: Confocal images showing the localization of endogenously expressed GFP::CHDP-1 (left), myr-mCherry (middle), and overlay (right) in the embryonic stages. Right: Normalized intensity of GFP::CHDP-1 and myr-mCherry around the cell membrane. Scale bar: 20 μm. (B) Left: Confocal images showing the localization of endogenously expressed GFP::CHDP-1 (left), myr-mCherry (middle), and overlay (right) in the PVD cell body. Right: Normalized intensity of GFP::CHDP-1 and myr-mCherry around the PVD cell body membrane. Scale bar: 5 μm.
(TIFF)

**S6 Fig. CHDP-1 is required for actin assembly in the 4° dendritic growth cones in early L4 stage.** (A) Left: Confocal images of GFP::moesinABD in *wild-type* (upper), *chdp-1(tm4947)* mutant (lower) at the early L4 stage. Scale bar: 10 μm. Right: Confocal images from time-lapse movies of GFP::moesinABD in *wild-type* (upper) and *chdp-1(tm4947)* mutant (lower) during the early L4 stage. Arrows: growth cones in *wild-type*, arrowheads: growth cones in *chdp-1 (tm4947)*. Scale bar: 5 μm. (B) Quantification of the average growth cones area labeled by GFP::moesinABD, and the ratio of the intensity of the growth cones in the 4° branches to that of the primary dendrite in *wild-type* and *chdp-1(tm4947)* during the early L4 stage. n = 100 growth cones for each group.
(TIFF)

**S7 Fig. CHDP-1 is required for actin assembly during dendrite formation.** (A) Left: Confocal images of Lifeact::GFP in *wild-type* (upper) and *chdp-1(tm4947)* mutant (lower) at early L3 stage. Scale bar: 10 μm. Right: Confocal images from time-lapse movies showing actin assembly in *wild-type* (upper) and *chdp-1(tm4947)* mutant (lower) during the early L3 stage. Arrows: growth cones of 2° branches labeled by Lifeact::GFP in *wild-type*, arrowheads: growth cones in *chdp-1(tm4947)*. Scale bar: 5 μm. (B) Quantification of the average growth cones area, and the ratio of the intensity in the growth cones of the 2° branches to that of the primary dendrite in *wild-type* and *chdp-1(tm4947)* during the early L3 stage. n = 100 growth cones for each group. (C) Left: Confocal images of Lifeact::GFP in *wild-type* (upper) and *chdp-1(tm4947)* mutant (lower) at early L4 stage. Scale bar: 10 μm. Right: Confocal images from time-lapse movies showing actin assembly in *wild-type* (upper) and *chdp-1(tm4947)* mutant (lower) during the early L4 stage. Arrows: growth cones of 4° branches labeled by Lifeact::GFP in *wild-type*, arrowheads: growth cones in *chdp-1(tm4947)*. Scale bar: 5 μm. (D) Quantification of the average growth cones area, and the ratio of the intensity in the growth cones of the 4° branches to that of the primary dendrite in *wild-type* and *chdp-1(tm4947)* during the early L4 stage. n = 100 growth cones for each group.
(TIFF)

**S8 Fig. Distribution of the NLP-12-positive dense-core vesicles in the primary dendrites.** (A) Confocal images of PVD dendrites (left), NLP-12 (NLP-12::Venus) (middle), overlay (right) in *wild-type* (upper), *chdp-1(tm4947)* mutant (lower). Arrows: NLP-12::Venus-positive dense-core vesicles in primary dendrites. Scale bar: 10 μm. (B) Quantification of NLP-12:: Venus fluorescence intensity in primary dendrites in a 100 μm area anterior to the PVD cell body (normalized to WT). Error bars: SEM. ***p < 0.001 by Student's t test. n = 20–30 for each genotype.
(TIFF)

**S9 Fig. ER and mitochondria transport into the high-ordered branches is defective in *chdp-1* mutant animals.** (A) Confocal images of PVD neuron (left), ER (labeled using a mCherry::SP12 reporter) (middle) and overlay (right) in *wild-type* (upper) and *chdp-1 (tm4947)* mutant (lower). Arrows: ER in the high-ordered branches. Scale bar: 20 μm. (B) Quantification of the number of 2° branches with ER invasion in a 100 μm area anterior to the PVD cell body. Error bars: SEM. ***p < 0.001 by Student's t-test. n = 20–30 for each genotype. (C) Confocal images from time-lapse movies showing transport of ER (upper) and overlay of ER (red) and PVD neuron (green) in *wild-type* (left) and *chdp-1(tm4947)* mutant (right) during early L3 stage. Arrows: ER in high-ordered branches. Scale bar: 5 μm. (D) Confocal images of PVD neuron (left), mitochondria (TOMM-20 1-54AA::GFP) (middle), overlay (right) in *wild-type* (upper) and *chdp-1(tm4947)* mutant (lower). Arrows: mitochondria in high-ordered branches. Scale bar: 20 μm. (E) Quantification of the number mitochondria in the high-

ordered dendrites in a 200 μm area anterior to PVD cell body. Error bars: SEM. ***p < 0.001 by Student's t test. n = 20–30 for each genotype. (F) Confocal images from time-lapse movies showing transport of mitochondria in PVD dendrites in *wild-type* (left) and *chdp-1(tm4947)* mutant (right) during L4 stage. Arrows: mitochondria in high-ordered branches. Scale bar: 5 μm.
(TIFF)

**S10 Fig. Transport of DMA-1::GFP vesicles in *wild-type* and *chdp-1* mutants.** (A) Upper: Confocal images of DMA-1::GFP in *wild-type* (left) and *chdp-1(tm4947)* mutant (right). Scale bar: 20 μm. Lower: Confocal images from time-lapse movies showing transport of DMA-1:: GFP vesicles in *wild-type* (left) and *chdp-1(tm4947)* mutant (right) at adult stages. Arrows: DMA-1::GFP vesicles moving in high-ordered branches. Scale bar: 5 μm. (B) Quantification of the number DMA-1::GFP vesicular units in the high-ordered dendrites in a 100 μm area anterior to the PVD cell body. Note that a DMA-1::GFP vesicular unit is defined as a punctum that is at least 2-fold brighter than the diffused signal along the dendrites. Error bars, SEM. ***p < 0.001 by Student's t-test. n = 20–30 for each genotype. (C) Upper: Confocal images showing the distribution of DMA-1::GFP vesicular units in *exoc-8 (ok2523)* mutant background in *wild-type* (left) and *chdp-1(tm4947)* mutant (right). Scale bar: 20 μm. Lower: Confocal images from time-lapse movies showing transport of DMA-1::GFP vesicles in *exoc-8 (ok2523)* mutant background in *wild-type* (left) and *chdp-1(tm4947)* mutant (right) in adult. Arrows: DMA-1::GFP vesicles in the high-ordered branches. Note that loss of *exoc-8* disrupts fusion between vesicles and the dendritic membranes, which make it easier to compare vesicular transport in *wild-type* and *chdp-1* mutant animals. Scale bar: 5 μm. (D) Quantification of the number DMA-1::GFP vesicular units in the high-ordered dendrites in a 100 μm area anterior to the PVD cell body in *exoc-8 (ok2523)* background. Error bars, SEM. ***p < 0.001 by Student's t-test. n = 20–30 for each genotype.
(TIFF)

**S11 Fig. Microtubule assembly in the growth cones of the anterior primary dendrite in WT and *chdp-1* mutants.** (A) A cartoon showing the area imaged to analyze the microtubule dynamics in the growth cone of the anterior primary dendrite. Scale bar: 2 μm. (B) Kymographs of EBP-2::GFP in a growth cone of the anterior primary dendrite in *wild-type* (left), *chdp-1(tm4947)* (right) during mid to late L2 stage. Scale bar: 2 μm. (C) Quantification of the number of EBP-2::GFP comets either moving away from the cell body (anterograde) or towards the cell body (retrograde) in *wild-type* and *chdp-1(tm4947)* mutant animals. Error bars, SEM. ns: non-significant by one-way ANOVA with the Tukey correction. n = 10 worms for each genotype. (D) Quantification of the length of tracks of the EBP-2 comets in *wild-type* and *chdp-1(tm4947)* mutant animals. Error bars, SEM. Ns: non-significant by one-way ANOVA with the Tukey correction. n = 10 worms for each genotype. (E) Quantification of the growth duration of tracks of the EBP-2 comets in *wild-type* and *chdp-1(tm4947)* mutant animals. Error bars, SEM. ns: non-significant by one-way ANOVA with the Tukey correction. n = 10 worms for each genotype. (F) Quantification of MT pause frequency in *wild-type* and *chdp-1(tm4947)* mutant animals. Error bars, SEM. ns: non-significant by one-way ANOVA with the Tukey correction. n = 10 worms for each genotype. (G) Left: Confocal images of PVD (top) dendrites and (down) axon (left), RAB-3 (middle), overlay (right) in *wild-type* (upper), *chdp-1(tm4947)* mutant (lower). Right: A cartoon showing the localization of RAB-3 in PVD dendrites and axon in *wild-type* and *chdp-1(tm4947)* mutants. Scale bar: 10 μm.
(TIFF)

**S12 Fig. Loss of *ced-10* does not phenocopy *chdp-1* mutants in PVD dendrite development.** (A) Confocal images showing the PVD morphologies of *wild-type*, *chdp-1(tm4947)* and *ced-10 (n3246)* mutant animals at 1DOA stage. Scale bar: 10 μm. (B-C) Quantification of (B) the number of $4^o$ branches in a 100 μm area anterior to the PVD cell body, and (C) the ratio of the intensity in $2^o$ branches to that of $1^o$ dendrites for the genotypes indicated. Error bars, SEM. ***$p < 0.001$ by one-way ANOVA with the Tukey correction. ns: non-significant. n = 20–30 for each genotype. (D) Confocal images of PVD dendrites (left), ER (labeled using a mCherry:: SP12 reporter) (middle), overlay (right) in *ced-10(n3246)* mutant. Arrows: ER in high-ordered branches. Scale bar: 20 μm. (E) Quantification of the number of 2° branches with ER invasion in *wild-type*, *chdp-1(tm4947)*, *ced-10(n3246)* mutant in a 100 μm area anterior to the PVD cell body. Error bars, SEM. ***$p < 0.001$ by one-way ANOVA with the Tukey correction. ns: nonsignificant. n = 20–30 for each genotype.
(TIFF)

**S13 Fig. Loss of *sax-1* restores the proprioceptive function of the PVD neurons in *chdp-1* mutants.** (A) Confocal images showing the PVD dendrite morphology of *wild-type*, *chdp-1 (tm4947)* and *chdp-1(tm4947); sax-1(PVD KO)* mutant animals at the 1-day-old adult stage. Scale bar: 20 μm. (B-C) Quantification of (B) the number of $2^o$, $3^o$ and $4^o$ branches, and (C) the ratio of the intensity of the $2^o$ branches to that of the primary dendrites in *wild-type*, *chdp-1(tm4947)* and *chdp-1(tm4947); sax-1(PVD KO)* in the 100 μm area anterior to PVD cell body. Error bars, SEM. *$p < 0.05$, **$p < 0.01$, ***$p < 0.001$ by one-way ANOVA with the Tukey correction. ns: non-significant. n = 20–40 for each genotype. (D) The representative moving tracks in *wild-type*, *chdp-1(tm4947)*, *sax-1(ky491)* and *chdp-1(tm4947); sax-1(ky491)* mutants. Scale bar: 200 μm. (E-G) Quantification of the (E) body length, (F) amplitude and (G) wavelength of tracks in late L4 for the genotypes indicated. Error bars, SEM. ***$p < 0.001$ by oneway ANOVA with the Tukey correction. ns: non-significant. n = 10 worms for each genotype.
(TIFF)

**S1 Table. Strains and plasmids used in this study.**
(DOCX)

**S2 Table. Source data file for quantifications.**
(XLSX)

## Acknowledgments

We thank Drs. David R. Sherwood and Li Tao for comments on the manuscript, Drs. Kang Shen, David R. Sherwood, Erik Jorgensen and Suhong Xu for reagents, and Drs. Xueliang Zhu, Kang Shen, Qiming Sun, Zhiping Wang, Heng Xu, Yu Feng and Hui Xiao for valuable discussion. Some strains were provided by the CGC, which is funded by NIH Office of Research Infrastructure Programs (P40 OD010440), and the MITANI Lab through the National Bio-Resource Project of the MEXT, Japan.

## Author Contributions

**Conceptualization:** Mei Ding, Wei Zou.

**Formal analysis:** Ting Zhao, Liying Guan, Xuehua Ma.

**Funding acquisition:** Baohui Chen, Wei Zou.

**Investigation:** Ting Zhao, Liying Guan, Xuehua Ma.

**Project administration:** Baohui Chen, Mei Ding, Wei Zou.

**Supervision:** Baohui Chen, Mei Ding, Wei Zou.

**Visualization:** Wei Zou.

**Writing – original draft:** Ting Zhao, Mei Ding, Wei Zou.

**Writing – review & editing:** Ting Zhao, Mei Ding, Wei Zou.

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
