## [Decision Letter · Decision Letter 0]

4 May 2022

Dear Dr Zou,

Thank you very much for submitting your Research Article entitled 'CHDP-1-dependent cortical actin assembly is required for dendritic development and transport in C. elegans neurons' to PLOS Genetics.

The manuscript was fully evaluated at the editorial level and by independent peer reviewers. The reviewers appreciated the attention to an important problem, but raised some substantial concerns about the current manuscript. Based on the reviews, we will not be able to accept this version of the manuscript, but we would be willing to review a much-revised version. We cannot, of course, promise publication at that time.

As you will see, reviewers have raised some concerns that can be addressed by clarifying the presentation, and others that may require further experimental work. Although some suggested approaches such as super-resolution microscopy may not be technically feasible, other experiments may be appropriate within the timeframe for a revision.

If you decide to revise the manuscript for further consideration at PLOS Genetics, please aim to resubmit within the next 60 days, unless it will take extra time to address the concerns of the reviewers, in which case we would appreciate an expected resubmission date by email to plosgenetics@plos.org.

[LINK]

We are sorry that we cannot be more positive about your manuscript at this stage. Please do not hesitate to contact us if you have any concerns or questions.

Yours sincerely,

Andrew D. Chisholm

Associate Editor

PLOS Genetics

Gregory P. Copenhaver

Editor-in-Chief

PLOS Genetics

Reviewer's Responses to Questions

**Comments to the Authors:**

Reviewer #1: Review PGENETICS-D-22-00369 (Zao…. Wei Zou)

Zhao et al. described the role of the cell cortex protein CHDP-1 in shaping the dendritic architecture of the C. elegans multidendritic neuron PVD. Through extensive genetic and imaging investigations, they showed that CHDP-1 controls the morphology of higher order PVD dendrites, promotes actin and microtubule assembly and facilitates the transport of the NLP-12 neuropeptide as well as organelles such as mitochondria and ER. All these likely translate into maintaining an intact PVD function for proprioception, as the chdp-1 mutant displays locomotion features of defective proprioception, including shallower and shorter waveforms in its sinusoidal movement. Interestingly, mutations in the sax-1 kinase gene suppressed the defects in PVD dendrite morphology, actin assembly and proprioceptive functions of the chdp-1 mutant, suggesting that SAX-1 antagonizes CHDP-1 functions. In general, the experiments were well planned and rigorously executed, and most of the results were convincing. Below I list a few concerns that the authors should address before the paper can be considered for publication in PLoS Genetics.

Major comments:

1. A key point in the paper is the localization of CHDP-1 to the cell cortex. However, the regular confocal microscopy used in Figure 4 is unlikely to provide the spatial resolution required to document CDHP-1 localization to the cell cortex. The authors should attempt STED super-resolution microscopy of GFP::CHDP-1 with an appropriate membrane marker such as myristoylated mCherry.

2. Figure S3 should be included in the formal text as domain-function analysis is important for understanding the molecular functions of CHDP-1. Line 192-193: CHDP-1 lacking the CH domain shows punctate signals in the cytosol and CHDP-1 without the helix domain shows diffuse cytosolic distribution. These changes should be explicitly described and discussed in the Results.

3. Line 149: The coverage index is a bit misleading, as it quantifies the length of the anterior primary dendrite, a phenotype that is distinct from that of the higher order dendrites described in Figures 1 and 2. It will be better to quantify the number of higher order dendrites in adulthood during the course of aging, as this is more coherent with the developmental phenotypes of the chdp-1 mutant. One could actually see that even in the wild type, the peripheral PVD dendrites deteriorate during aging, consistent with a prior report (Lezi et a., Neuron 2018).

4. Line 202-203: The authors may want to briefly explain how NATF works to detect the endogenous expression pattern of CHDP-1. In Figure 5B, the fluorescent signals of GFP::CHDP-1 are diffuse in the PVD dendrites. One cannot conclude that CHDP-1 and moesinABD colocalize on the basis of such diffusely distributed signals, and I do not think this supports the statement that CHDP-1 is enriched in the growth cone. The seven consecutive GFP(11) modules attached to the CHDP-1 protein could be a source of technical concern. The authors should address this issue with alternative methods.

5. Figure 5D and 5F: It seems to the reviewer that at L3, the dendrite growth cones in the wild type display extended palm-shaped morphology and undergo dynamic changes in their appearance, whereas those in the chdp-1 mutants are thin, pointed and less dynamic. The current quantification of the growth cone area does not highlight the dynamic nature of the growth cone morphology. Did the authors quantify the same 100 growth cones in respective genotypes? This is hard to imagine, as not all growth cones are present at any given time points during the recording: some are yet to appear while others are retracting. The authors need to provide more detailed account of their time-lapse imaging methods and characterize the dynamic nature of the dendrite growth cones.

6. Line 235-238: The authors imply that NLP-12 is secreted from the peripheral PVD dendrites, which is quite unusual. They should also examine whether NLP-12::Venus in the PVD axon is also altered in the chdp-1 mutant. NLP-12::Venus puncta in the peripheral PVD dendrites are sparse (Figure 7A, 7B, 7D and 7E), with a mean density of one puncta per 100 um of dendrite length. This seems to be a pretty low number for something that is so important for proprioception and neuromuscular function. Figure 7C and 7D were poorly explained in the text. Please explain the design of the GBP assay and the interpretation of the data. In Figure 7D, does the “wild type” animal also express Pdpy-7::GBP::SAX-7, as in the sax-7 and chdp-1 mutants? The label of Pdpy-7::GBP::SAX-7 is inconsistent between 7D and 7E.

7. Line 243-244: To show that NLP-12 dendritic transport is disrupted in the chdp-1 mutant, the authors should quantify NLP-12::Venus puncta in the primary PVD dendrite and/or measure NLP-12::Venus movement by kymograph analysis.

8. It is unclear how CHDP-1, which the authors claim to be involved in the assembly of cortical actin near the cell membrane, is required for vesicle transport and trafficking, as actin species for this transport function is distinct from that underneath the cell cortex. The authors should offer some speculation.

9. Figure 8: The data showed that both plus end-out and minus end-out microtubules (MTs) exist in the secondary PVD dendrites, based on the EBP-2 comet patterns. In the chdp-1 mutant, all EBP-2 comets were absent, raising an intriguing possibility that CHDP-1 is required for the assembly of both plus end-out and minus end-out MTs. Given that CHDP-1 is a cell cortex protein, how does it regulate the assembly of MTs, especially for MTs of opposite polarity? Although in Discussion (Line 343-344), the authors favored the model that CHDP-1 regulates MTs indirectly via actin assembly, there is no evidence for this. The authors may want to explore whether CHDP-1 is associated with the microtubule cytoskeleton.

Minor comments:

1. The Materials and Methods section should be significantly substantiated. Many experiments are inadequately explained, preventing the readers to grab essential details of the experiments. This is particularly evident with the split-GFP assay for cell-specific detection of CHDP-1 expression and the local neuropeptide secretion assays.

2. Figure 1D: What is the orientation of the STED images of secondary dendrites here?

3. The genotype label chdp-1(tm4947) in all the figures should be italicized. Please carefully check this.

4. Line 129: Change chdp-1-/- to chdp-1(tm4947) throughout the paper, and indicate that tm4947 is a representative null, deletion allele where it first appears in the manuscript.

5. Line 130-133, 137-140: There is no need to describe the quantitative results in numerical details, as these could be read from the respective figures directly. Please simplify tis and remove redundant descriptions throughout the paper.

6. Figure 2A and 2F: Please show snapshots of time-lapse imaging that have both growing and retracting secondary dendrites. There are only growing dendrites in the current figure.

7. Figure 6: The dendrite defects of the chdp-1 mutant are “relatively mild” compared to those in the dma-1 mutant. However, both display similar proprioceptive defects quantified by the amplitude and wavelength of the sinusoidal movements on the agar. The authors should offer some explanation.

8. Line 234: Please change “why” to “how”. As scientists, we deal with HOW things happen rather than WHY they happen.

9. Line 247: It is endoplasmic “reticulum”, not “reticulon”.

10. Figure 7F: The signal of ER labeled by SP12 is continuous and diffuse in the primary, secondary and some of the tertiary PVD dendrites. I would have expected a puncta appearance, as other subcellular compartments in the neuronal process. Could the authors explain this peculiar pattern of dendritic ER or confirm it with another ER marker such as KDEL?

11. Figure S5A: There seems to be a diffuse background GFP signal in addition to the more punctate DMA-1::GFP. It is not easy to appreciate the presence of the punctate signals. The authors need to provide their criteria of quantifying DMA-1::GFP vesicles. Please explain the exoc-8 genetic background in the main text.

12. Line 303: The statement that SAX-1 is a negative regulator of actin assembly is based on its antagonism of CHDP-1. However, direct evidence in support of this conclusion is missing in the current study. The authors are suggested to tone it down or clearly indicate that this is a speculation.

Reviewer #2: The manuscript by Zhao et al. describes the role for the calponin homology containing protein, CHDP-1 in dendritic morphogenesis of the nociceptive receptor neuron, PVD in C. elegans. CHDP-1 was previously shown by the authors to be associated with the CED-10/RAC-1 to regulate actin cytoskeleton in BDU & PLM neuron. In this study, they extend the functional analysis of CHDP-1 to PVD morphogenesis. Through a set of experiments, they show that CHDP-1 function is required for proper dendritic arborization. CHDP-1 function in the PVD to modulate actin assembly, distribution of organelles and microtubules and proprioception. The authors suggest that CHDP-1 regulates actin assembly which in turn affects the intracellular cytoskeleton and that this is necessary for the development, maintenance, and function of the PVD dendrites. This an interesting paper with several well executed experiments and adds a new player into the mix of components that is required for the dendritic patterning in the PVD.

In general, I found the paper to be very dense with a lot of different experiments testing various hypotheses. Some of the data are strongly supported by the evidence provided by the authors however a few I find to be preliminary. It would be useful if the authors could distil the information in the main figures by only including the data that shows the strongest evidence. The main takeaway for me is that chdp-1 is involved in proper organization of the cytoskeleton and organelles in the higher order branches. This is a cell autonomous function and that this requires the CH and the helix within chdp-1. Lack of chdp-1is leads to dendrite degeneration and defective proprioception due to impaired branches. This function may not be through the ced-10 protein but is suppressed by the sax-1 kinase which can affect the actin protrusion dynamics arguing that chdp-1 is affecting an actin dependent process.

Specific comments.

1. Results for chdp-1 mutant effects on 2º branches are confusing. Figure 1B shows an increase of 2º branches in mutants of chdp-1 vs. WT. Representative images (Figure 1A), however, appear to show similar 2º branches in all three panels. It is that clear chdp-1 mutants affect the 2º, 3º & 4º branching dynamics/frequency based on the time lapse data. The strongest phenotype lies in the branching defects of higher order dendrites. Given the penetrance of that phenotype it, focusing on the 3º & 4º branching defects alone would be sufficient to do the functional analysis done with the CHDP-1 protein. Also, Figures 1D, E & F which shows the width of the primary dendrite is not adding much as it is difficult to know the exact parameters that is being measured as well as the reason for swelled 1º dendrite.

2. In Fig5 the authors very nicely use a splitgfp system to visualize CHDP-1 exclusively in PVD. They also use mCherry:moesinABD as a growth cone marker and states that “GFP7x::CHDP-1 was enriched in the dendritic growth cones, and colocalized with the F-actin 205 probe mCherry::moesin actin binding domain (moesinABD)”. I am struggling to see the enrichment of CHDP-1 in the growth cones. From the images in Fig.5A it does look like the CHDP-1 is uniform. Would it be possible to use a different counter marker such as membrane to support that CHDP-1 is indeed enriched in growth cone? An alternate possibility is to look at the growth cones of 1º dendrite which are much larger in size and is also much more defined.

Minor comments/suggestion

Figures 1 & 2 could be combined and some of the data from figure 1 (STED & 2º branch quantification) could be moved to supplemental data section. The data in on higher order branching and the time lapses are clearly strong to support chdp-1 mutant phenotype. Also, it might be worth combining the dendritic regeneration and proprioception data as it is a functional consequence to the neuron.

Reviewer #3: The manuscript by Zhao et al describes the function of CHDP-1 in PVD neuron development. The mutant shows some interesting phenotype: there are ectopic 2o and 3o dendrites, however, less 4o dendrite. The 4o dendrites are not abolished though like other F-actin cytoskeleton related mutants such as hpo-30, tiam-1, act-4, dma-1. The phenotype seems to be more specific to ectopic 2o branches emerging from the primary one. Although it is an interesting study and it can be published in PLOS-GENETICS. However, several questions remain in terms of how CHDP-1 regulates the cytoskeleton element in the PVD neuron.

1) It is not clear how they conclude CHDP-1 affects cortical actin cytoskeleton specifically. It may affect cytosolic F-actin structures as well. It seems in general F-actin cytoskeleton is dim in the mutant dendrites.

2) It is not clear whether CHDP-1 promotes assembly or disassembly of F-actin. Some photo-bleaching experiment followed by recovery could tell how F-actin dynamics is affected.

3) Can it affect cross-linking of F-actin through non-muscle myosin? I think both drugs and mutants are available to test this

4) It is also not clear how it affects microtubule cytoskeleton. The primary effect could be microtubule cytoskeleton as well.

5) Related to this question: The ectopic 2o dendrites could be due to the over stabilization of microtubules. Or is it due to the instability of microtubules. Treating the mutant with colchicine or taxol might shed some light.

6) Thickening of the primary dendrites could be a sign of abnormal microtubule cytoskeleton. Although according to the Supple Figure 6, authors conclude that EBP-2 tracks in primary dendrites are unaffected, I recommend to show a better kymograph from the mutant. It looks like there is some movement during imaging. I also suggest to look at the growth duration and length of these tracks in wild type and mutant. Also check the pause frequency. It might tell whether there is any effect on growth or depolymerization.

7) Author may discuss the possibility that CHDP-1 can affect microtubule cytoskeleton directly in an unknown mechanism. Since it is not tested yet whether it associates with microtubule or not, this possibility always remains. Although I am not asking to do more experiments on this.

8) Did they check the synaptic vesicles in the mutant dendrites? RAB-3 reporters?

9) The connection with SAX-1 is also not clear. Activity or amount of SAX-1 might be upregulated in chdp-1 mutant? Is there any SAX-1 reporter available to test? If it is possible, it can be addressed.

**Have all data underlying the figures and results presented in the manuscript been provided?**

Reviewer #1: Yes

Reviewer #2: None

Reviewer #3: Yes

PLOS authors have the option to publish the peer review history of their article (what does this mean?). If published, this will include your full peer review and any attached files.

Reviewer #1: No

Reviewer #2: No

Reviewer #3: No

---

## [Decision Letter · Decision Letter 1]

17 Aug 2022

Dear Dr Zou,

We are pleased to inform you that your manuscript entitled "The cell cortex-localized protein CHDP-1 is required for dendritic development and transport in C. elegans neurons" has been editorially accepted for publication in PLOS Genetics. Congratulations!

Please note that Reviewer #1 has some minor suggestions (see below) that you can attend to as you prepare the final draft of your manuscript for the production team (the editorial team will not need to re-evaluate).

Yours sincerely,

Andrew D. Chisholm

Academic Editor

PLOS Genetics

Gregory P. Copenhaver

Editor-in-Chief

PLOS Genetics

Comments from the reviewers (if applicable):

Reviewer's Responses to Questions

**Comments to the Authors:**

Reviewer #1: The authors had addressed my comments nicely with additional experiments and revision of the manuscript. The work is significantly improved after these revisions and could be considered for publication at PLoS Genetics. I have a few minor suggestions for the authors before they finalize the paper, as listed below.

Figures 1C/2E/2H/3C/6G/7C: It will be more intuitive to present the ratio of myr-mCherry fluorescence or TBA-1 intensity between the primary and secondary branches as secondary/primary, which will make the values for the chdp-1 mutants lower than those for the wild type. The only such example is Fig.7E (LifeAct).

Manuscript typos and errors: for example,

Line 143: "were" should be "was"...

Line 250/259: reference #27 seems to be "Tao et al.", not "Li et al."?

Please carefully check the manuscript again and correct all these errors. In particular, panel labels for the supplemental figures should be specified, such as "S3A Fig".

Reviewer #2: In the revised manuscript, the authors have satisfactorily addressed my concerns, and I recommend publication.

Reviewer #3: The authors have addressed all my comments and it is significantly improved.

I recommend publication.

**Have all data underlying the figures and results presented in the manuscript been provided?**

Reviewer #1: Yes

Reviewer #2: **No: **This is not necessary

Reviewer #3: Yes

PLOS authors have the option to publish the peer review history of their article (what does this mean?). If published, this will include your full peer review and any attached files.

Reviewer #1: No

Reviewer #2: No

Reviewer #3: No

**Data Deposition**

http://datadryad.org/submit?journalID=pgenetics&manu=PGENETICS-D-22-00369R1

**Press Queries**

---

## [Editor Report · Acceptance letter]

15 Sep 2022

PGENETICS-D-22-00369R1 

The cell cortex-localized protein CHDP-1 is required for dendritic development and transport in C. elegans neurons 

Dear Dr Zou, 

We are pleased to inform you that your manuscript entitled "The cell cortex-localized protein CHDP-1 is required for dendritic development and transport in C. elegans neurons" has been formally accepted for publication in PLOS Genetics! Your manuscript is now with our production department and you will be notified of the publication date in due course.

With kind regards,

Anita Estes

PLOS Genetics

On behalf of:
